# Establishment and maintenance of *NRT2.1* inter-individual variability in plants

**Charlotte Lecuyer[1], Alexandre Vettor[1], Cécile Fizames[1], Hélène Javot[2], Antoine Martin[1], Mona Mazouzi[1], Marie-Hélène Montané[2], Sandra Cortijo[1]***

**1** IPSiM, Univ Montpellier, CNRS, INRAE, Institut Agro, Montpellier, France, **2** Aix Marseille Univ, CEA, CNRS, BIAM, Marseille, France

☙ These authors contributed equally to this work.
* sandra.cortijo@cnrs.fr

## Abstract

Morphological phenotype and gene expression differences are observed between genetically identical plants grown in the same environment. While we now have a good understanding of the source and consequences of transcriptional differences observed between cells, our knowledge is still very limited regarding variability between multicellular organisms. We characterised this variability using the high-affinity nitrate transporter gene *NRT2.1* as a model for high inter-individual transcriptional variability. Thanks to a combination of live imaging and transcriptomics, we show that the differences in expression of this gene between plants are established in young seedlings and maintained for up to three weeks. However, the expression level of *NRT2.1* in plants does not permit predicting its expression in the next generation. Our results also indicate that these expression differences could have phenotypic consequences on root growth and nitrate uptake mediated by *NRT2.1*. Finally, we observed enriched photosynthesis-related functions among genes whose expression correlates with *NRT2.1* in individual seedlings. Our study thus demonstrates that a global coordination of the genes involved in the carbon/nitrogen (C/N) balance in plants is established in young seedlings, at different levels in each plant, and maintained over time. Our results also highlight the fact that not all transcriptional regulators of *NRT2.1* were identified, and propose UNE10 as a transcription factor for further study focused on its possible involvement in this pathway. This work shows that thanks to single-plant analysis of gene expression, we can gain new knowledge on the mechanisms behind a phenotype of interest that is normally masked in studies performed on pooled plants.

**Data availability statement:** RNA-seq data are available on ArrayExpress with the accession number E-MTAB-15039. Other raw data and scripts related used in the article can be found on GitHub (https://github.com/scortijo/Cortijo_NRT21_noise_paper_data_and_code) as well as on Recherche Data Gouv (https://doi.org/10.57745/RQMMFO).

**Funding:** This work was supported by grants from the I-SITE MUSE (AAP21REC-FRA02 to SC) and from the National Agency for Research (ANR-22-CE20-0020 to SC). AV received a salary from AAP21REC-FRA02 and ANR-22-CE20-0020 grants and CL received a salary from ANR-22-CE20-0020. The funders had no role in study design, data collection and analysis, decision to publish, or preparation of the manuscript.

**Competing interests:** The authors have declared that no competing interests exist.

## Author summary

Transcriptional differences between genetically identical plants could have an impact on the survival of plant populations. However, we still lack knowledge of how transcriptional variability between plants arises, and if it has an impact on morphological phenotypes. Using a combination of live imaging and transcriptomics, we show that a global coordination of genes involved in the carbon/nitrogen balance in plants is established at different levels in young seedlings and maintained over days. Our work also showcases the power of single-plant approaches in the study of variability, as well as understanding the mechanisms controlling a phenotype of interest.

## Introduction

Genetically identical plants grown in the same environmental conditions can display differences in morphological phenotypes and gene expression. This inter-individual variability has been observed in plants regarding seed germination timing [1,2], aerial phenotypes such as phyllotaxy [3], plant height, rosette size or number of flowers [4], and lateral root length [5]. The level of this variability is at least partially genetically controlled, as shown by several successful GWAS and QTL studies which identified genomic regions that can influence the level of inter-individual variability in the different genotypes [1,4,6–8]. It has also been shown that transcriptional inter-individual variability is widespread, with 9% of the transcriptome displaying a high level of variability [9]. The existence of this variability can have both positive and negative consequences, including for instance the suspected positive impact of seed germination variability on the survival of plant populations in the desert during unexpected droughts after a rainfall [2,10–12]. On the other hand, variability in a range of phenotypes such as seed germination or flowering time is undesirable in crops, when trying to optimize farming procedures. In this context, it is essential to understand the source of transcriptional variability in plants as well as its phenotypic consequences.

Although transcriptional noise at the single-cell level is now well documented, our understanding of inter-individual gene expression variability in multicellular organisms remains limited. It is known that the source of cell-to-cell transcriptional noise is the inherent stochasticity of the transcriptional process itself, with layers of factors that can either enhance or buffer this stochasticity [13,14]. These factors can result from genomic, epigenomic or regulatory circuit structures, as well as cell age and the global transcriptional state of the cell or its microenvironment [15–18]. Regarding differences in expression observed between multicellular organisms, identifying the source of this variability is more complex. Since multicellular organisms are composed of thousands of cells, it is not clear if the factors involved in the variability between cells also explain the variability between individuals. A transcriptomic study of inter-individual variability in *Arabidopsis thaliana* showed that highly variable genes have a genomic and epigenomic structure similar to highly variable genes from

unicellular organisms [9], suggesting that the same mechanisms are at play. However, much remains unknown about the sources and consequences of inter-individual variability in plants. In particular, it is not known if inter-individual variability depends on the plant growth stage, or if differences in expression between plants are constant over time. In addition, the dependence of this parameter on the environment needs to be evaluated.

In this study, we show that the high affinity nitrate transporter *NRT2.1* has a high inter-individual transcriptional variability in a range of assayed growth conditions. This variability in *NRT2.1* expression is correlated with differences between seedlings for several phenotypes such as root growth and the activity of NRT2.1 protein in importing nitrate. We found that expression differences between plants for this gene are established early on, shortly after the heterotrophic-to-autotrophic transition, and are maintained for at least three weeks but are not predictive of *NRT2.1* expression levels in the next generation. Finally, we observed that genes co-expressed with *NRT2.1* are involved in photosynthesis rather than in nitrate acquisition, highlighting the regulatory link between nitrate and carbon homeostasis in plants.

## Results

### Selection and validation of the nitrate transporter gene *NRT2.1* as a prime example for the study of highly variable expression between plants

In preparation of a detailed study of gene expression variability, a screen was performed to select a single highly variable gene. For this, we exploited a previously published RNA-seq dataset describing the variability of gene expression along a single day/night cycle, to identify genes that are highly variable at all time points [9]. In addition, such genes had to harbour genomic and epigenomic features identified for highly variable genes in this study, including a compact chromatin state and regulation by a high number of transcription factors [9]. Among the list of candidates, we focussed on two additional criteria: genes with a known luciferase promoter reporter line (permitting the long-term tracking of gene expression); and an extensive literature about their transcriptional and post-transcriptional regulation, as well as their regulatory factors. Based on these requirements, we selected *NRT2.1* (AT1G08090), which encodes a high-affinity nitrate transporter and has an extensive associated literature [19–21]. For comparison, we also selected two control genes (AT1G08930 and AT2G28810) from the previously mentioned study on inter-individual variability for the entire transcriptome of *Arabidopsis thaliana* [9] and in which their level of variability was validated by RT-qPCR. Both genes are expressed throughout the entire day/night cycle, and AT1G08930 is a highly variable gene (HVG) during the day, whereas AT2G28810 is a lowly variable gene (LVG) at all time points.

To further assess the level of plant-to-plant transcriptional variability for *NRT2.1*, as well as our HVG and LVG controls, we performed RT-qPCR on RNA extracted from individual seedlings grown on media containing a low (1 mM $KNO_3$) concentration of nitrate. As *NRT2.1* is expressed in roots, all experiments were performed on plants grown in Petri dishes for easy imaging and access to the roots. *NRT2.1* is known to be highly expressed in low-nitrate conditions, with an essential function in growth [22,23]. Gene expression in individual seedlings was quantified in independent studies, each involving 24–32 seedlings (with 8 seedlings per plate). The coefficient of variation (CV) was calculated based on the individual expression values, by dividing the standard deviation by the average expression level in the seedlings. The higher the CV, the higher the plant-to-plant transcriptional variability for the analysed gene. This experiment confirmed the variable nature of *NRT2.1* expression, since its CV was equivalent to the established HVG control. Its CV was also significantly higher than the LVG reference (Fig 1A). This result shows that *NRT2.1* expression is highly variable between plants, including in our growing conditions.

The high variability of *NRT2.1* expression was confirmed using a previously described luciferase *NRT2.1* promoter reporter line [24], by comparing its level of signal variability to that of a *p35S:LUC* line [25]. The cauliflower mosaic virus (CaMV) *35S* promoter allows for the constitutive high-expression of the luciferase reporter, making it a good control for low inter-individual variability (in the absence of any luciferase reporter lines for the LVG control AT2G28810). We found a

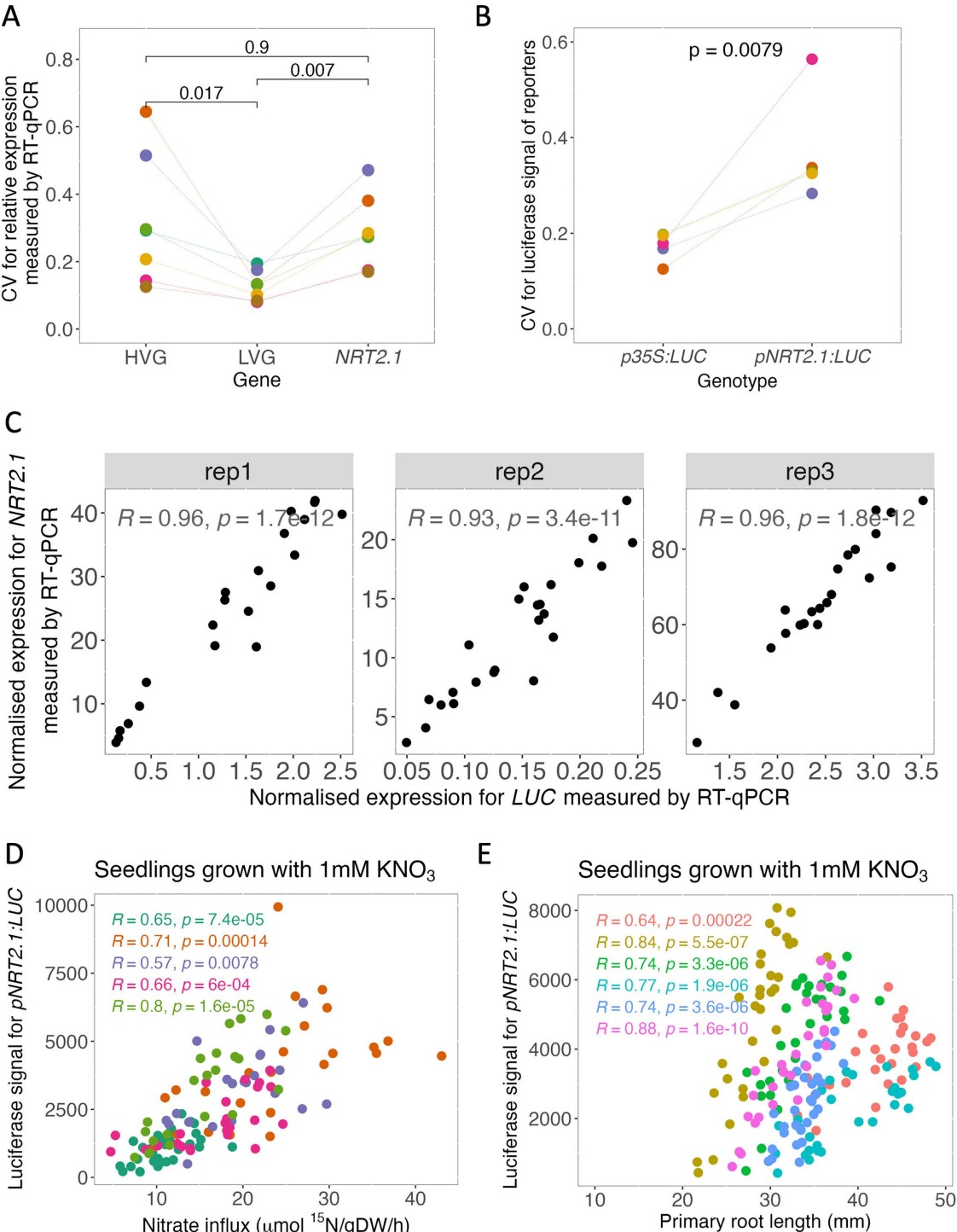

**Fig 1. *NRT2.1* expression is highly variable between plants and correlates with phenotypes. (A)** Inter-individual transcriptional variability measured by RT-qPCR for a highly variable gene (HVG, AT1G08930), a lowly variable gene (LVG, AT2G28810), and *NRT2.1*. The variability was measured using the CV (standard deviation/mean). Each colour represents an independent assay with 24 seedlings per assay grown on media with a low nitrate concentration (1 mM). Wilcoxon test results are included. **(B)** Measurement of inter-individual variability for the *pNRT2.1* and *p35S* promoters using the *pNRT2.1:LUC and p35S:LUC* luciferase reporter lines, respectively. Each colour represents an independent assay with at least 28 seedlings per assay grown on media with a low nitrate concentration (1 mM). The Wilcoxon test result is included. **(C)** Comparison of the quantity of mRNA measured by

RT-qPCR for the *NRT2.1* and luciferase genes in the *pNRT2.1:LUC* reporter line. Each point corresponds to one seedling and each plot to an independent assay. The Spearman correlation test results are included. **(D-E)** Comparison for plants grown in media with low (1 mM) nitrate concentration of the *pNRT2.1:LUC* signal and (D) the nitrate influx of HATS or (E) the primary root length. Each point corresponds to a single seedling and each colour to independent assays. A Spearman correlation test result for each assay is included.

higher inter-individual variability for the luminescence signal of the *NRT2.1* promoter compared to the *35S* promoter (Fig 1B). This result confirmed the high variability of *NRT2.1* between plants, by two distinct methods (RT-qPCR and luminescence reporter quantification of expression). It also showed that the *pNRT2.1:LUC* reporter line could be used to study *NRT2.1* variability, presenting significant advantages in terms of cost and time as compared to RT-qPCR. Before our final decision to use luminescence imaging to study the inter-individual transcriptional variability of *NRT2.1*, we compared by RT-qPCR the expression levels of the endogenous *NRT2.1* gene with the luciferase gene under control of the *NRT2.1* promoter in single seedlings. The strong positive correlation between the expression of the two genes (Fig 1C) indicated their co-regulation in single seedlings. The luminescence from the *pNRT2.1:LUC* reporter line can thus be used as a proxy for the expression of endogenous *NRT2.1*.

After this validation, we wanted to test whether the high inter-individual transcriptional variability of *NRT2.1* could be dependent on its overall expression level. *NRT2.1* expression level and its function are known to be influenced by the nitrate concentration in the plant environment [19,20,26]. We thus analysed the luminescence variability of *pNRT2.1:LUC* seedlings grown in either low (1 mM $KNO_3$) or high (10 mM $KNO_3$) nitrate concentrations. Neither of these two conditions (low or high nitrate) are perceived as a stress for the plant, as they stay within a physiological range. Based on 23 experiments (each containing around 25 seedlings per nitrate condition), *NRT2.1* expression was observed to be higher for plants grown in a low-nitrate concentration (S1A Fig), in agreement with the literature [19,20,26]. However, when plant-to-plant variability was compared instead of overall expression, we observed no statistically significant differences for the CV in low versus high nitrate concentrations (S1B Fig). This result shows that the *NRT2.1* expression level can change while maintaining its inter-individual variability.

### *NRT2.1* expression differences between plants correlate with different phenotypes

To evaluate the significance of *NRT2.1* inter-individual variability, *NRT2.1* expression differences between plants were compared with the amplitude of average changes in *NRT2.1* transcription caused by growth in distinct environments (S2A Fig). For the *pNRT2.1:LUC* reporter line, the difference in the average signal of seedlings grown in low (1 mM $KNO_3$) or high (10 mM $KNO_3$) nitrate conditions (statistically significant difference of 1.7x) was on the same order of magnitude as differences between seedlings in a given condition (1.4 to 1.9x between the 1st and the 3rd quartile, S2A Fig). The differences in expression between these two environmental conditions are known to have a physiological consequence, as NRT2.1 is the main nitrate transporter in low-nitrate conditions while other transporters, like NRT1.1, are involved in high-nitrate conditions [27,28]. In this context, the extent of *NRT2.1* transcriptional variability between plants seemed significant, with the level of differences between plants in one environment being at least as much as the transcriptional response of that gene to changes in nitrate concentrations.

To test whether *NRT2.1* variability between plants could be linked to phenotypic consequences, we scored the level of luminescence of individual *pNRT2.1:LUC* seedlings along with several phenotypes known to be affected in the *nrt2.1* mutant, including the activity of the high-affinity nitrate transport system (HATs), primary root length, and lateral root density [22,29–31]. First, we measured the luciferase signal for individual seedlings and then performed a nitrate influx experiment in the same seedling to measure the activity of HATs. Since NRT2.1 explains most of the transport observed for HATs when plants are in low-nitrate conditions [22,23], our measurement can be considered as a proxy for NRT2.1 protein activity. We observed a strong positive correlation between the luciferase signal for *pNRT2.1:LUC* and the nitrate

influx for HATs in all independent assays (5 total) when seedlings are grown on low-nitrate (1 mM $KNO_3$) conditions (Figs 1D, S2D, Spearman p-value < 0.05). However, this correlation was lost in all 4 replicates when seedlings are grown on high-nitrate (10 mM $KNO_3$) conditions (S2C, S2D Fig). This result suggests that *NRT2.1* transcriptional variability correlates with NRT2.1 transporter function variability, but strictly when the plants are grown in conditions where NRT2.1 is the main transporter.

To explore the possible impact of *NRT2.1* transcriptional variability on other macroscopic phenotypes representative of plant growth and development, we performed the same analysis for primary root length and lateral root density (number of lateral roots/primary root length). The primary root luminescence (reporting *NRT2.1* promoter activity) strongly correlated with the primary root length measured in individual seedlings for plants grown in low (1 mM $KNO_3$) as well as high (10 mM $KNO_3$) concentrations of nitrate (6 independent assays, Figs 1E and S2E). We also observed a positive correlation between primary root luminescence and lateral root density measured on single seedlings, although this was mainly for plants grown in a low (1 mM $KNO_3$) nitrate concentration (S2B Fig). For plants grown in a high (10 mM $KNO_3$) nitrate concentration, the correlation with the lateral root density was not as strong, with significance for only 4 out of 6 replicates (S2F Fig). In our control, we did not observe any correlation between the *p35S:LUC* signal and the primary root length in low- or high-nitrate concentrations, except in one of the three replicates (S2G, S2H Fig). This indicates that the correlation between *pNRT2.1:LUC* and primary root length is a distinct property of the *NRT2.1* promoter, but not of the constitutive *35S* promoter. This type of correlation must therefore be evaluated on a promoter to promoter basis, with *pNRT2.1* being a strong example of a positive correlation with primary root length.

These results suggest a possible phenotypic consequence of *NRT2.1* inter-individual transcriptional variability. Since the root phenotype of *nrt2.1* varies depending on the environmental conditions tested in the literature, we decided to measure the impact of the *nrt2.1* mutation on primary root length and lateral root density in our conditions. We found that the *nrt2.1* mutant had a shorter root length and a lower lateral root density than the Wild-Type (WT) when they are grown on low-nitrate conditions (1 mM $KNO_3$) (Fig 2A, 2B); this difference was seen in all three replicates, despite divergences in the absolute levels in the replicates (Fig 2). This is in agreement with the positive correlation we measured between these two root phenotypes and the luciferase signal for *pNRT2.1:LUC*, showing that plants with a high *NRT2.1* promoter activity have a longer primary root and a higher density of lateral roots. However, no reproducible difference between the two genotypes was observed when the plants were grown on high-nitrate conditions (10 mM $KNO_3$) (Fig 2B).

## Differences in expression between seedlings are established in young seedlings

To better characterise *NRT2.1* inter-individual variability, we next decided to define when the differences in expression between plants are established. For this, we followed *NRT2.1* promoter activity every 4 hours for 10 days from the exit of stratification, using the *pNRT2.1:LUC* line (S1 Movie). The *p35S:LUC* line was also analysed in parallel as a control. This was done using an automated plant growth chamber with controlled light phases for growth and dark phases with luminescence imaging, in which up to 11 plates can be analysed in the same experiment (Lumalum) [32]. Luminescence signals measured along the primary roots of *pNRT2.1:LUC* seedlings show a transient signal peak at germination, followed by a gradual increase in signal over several days (Fig 3A). At around seven to eight days, differences between seedlings increased in the *pNRT2.1:LUC* signal. This was very striking in some plates (see plates 1 and 5), with a signal that continued to increase in some seedlings (e.g., seedling 2 in Fig 3A, 3B), while it stagnated or even decreased in others (e.g., seedling 1 in Fig 3A, 3B). From this moment, seedlings usually maintained a relatively high or low *pNRT2.1:LUC* signal until the end of the experiment (Fig 3A, 3B). This is different from what was observed for the *p35S:LUC* line, where the signal increased gradually in a similar way in all seedlings (Fig 3A). This timeline and its trends were very reproducible, with similar observations made in independent assays (such as in S3 Fig). These results indicate that differences in *NRT2.1* expression between seedlings are established at around 7 days and are then maintained over time.

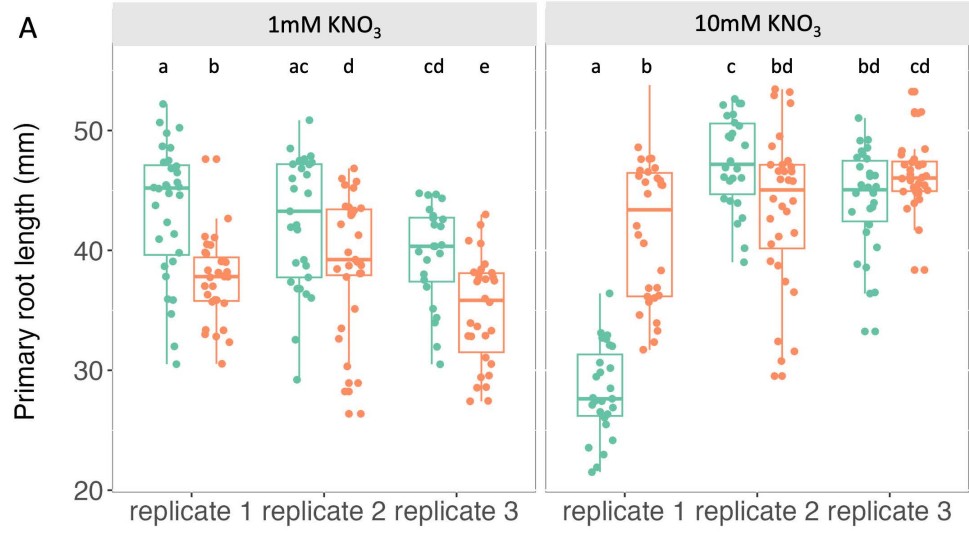

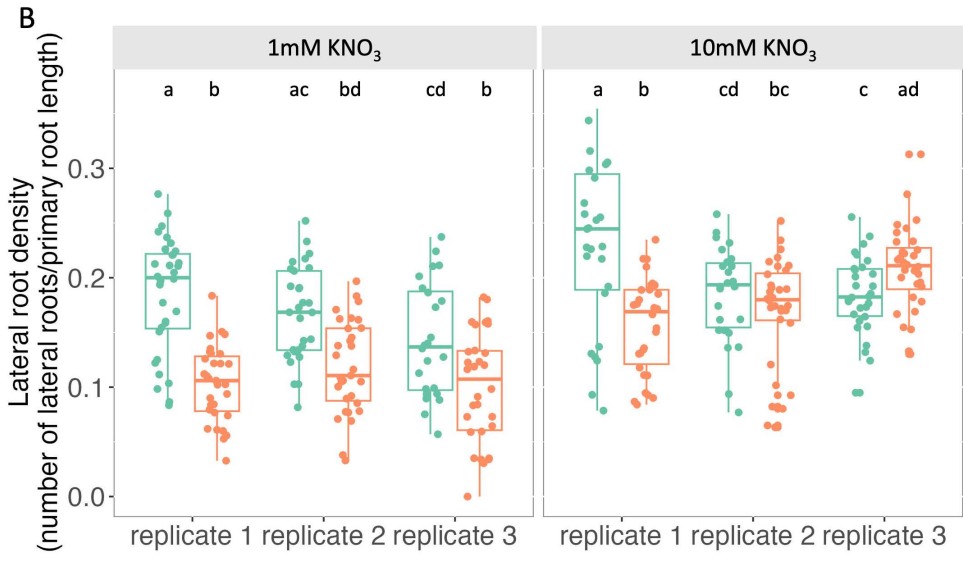

**Fig 2. Impact of the *nrt2.1* mutation on root phenotypes. (A-B)** Measurement of (A) the primary root length and (B) lateral root density (number of lateral roots/primary root length) for Col-0 WT and the *nrt2.1* mutant for plants grown with either 1 mM of $KNO_3$ (left) or 10 mM of $KNO_3$ (right) in their growth media. A total of 3 biological replicates are shown. Letters correspond to different groups identified doing Wilcoxon rank sum tests between all pairs of samples in a plot.

To test if differences in *NRT2.1* expression are established before or after the heterotrophic-to-autotrophic transition in young seedlings, we analysed the primary root growth of seedlings starting at germination. Primary root growth arrest was observed in some seedlings at around 3 days (Fig 3C), very likely due to a failure in the heterotrophic-to-autotrophic

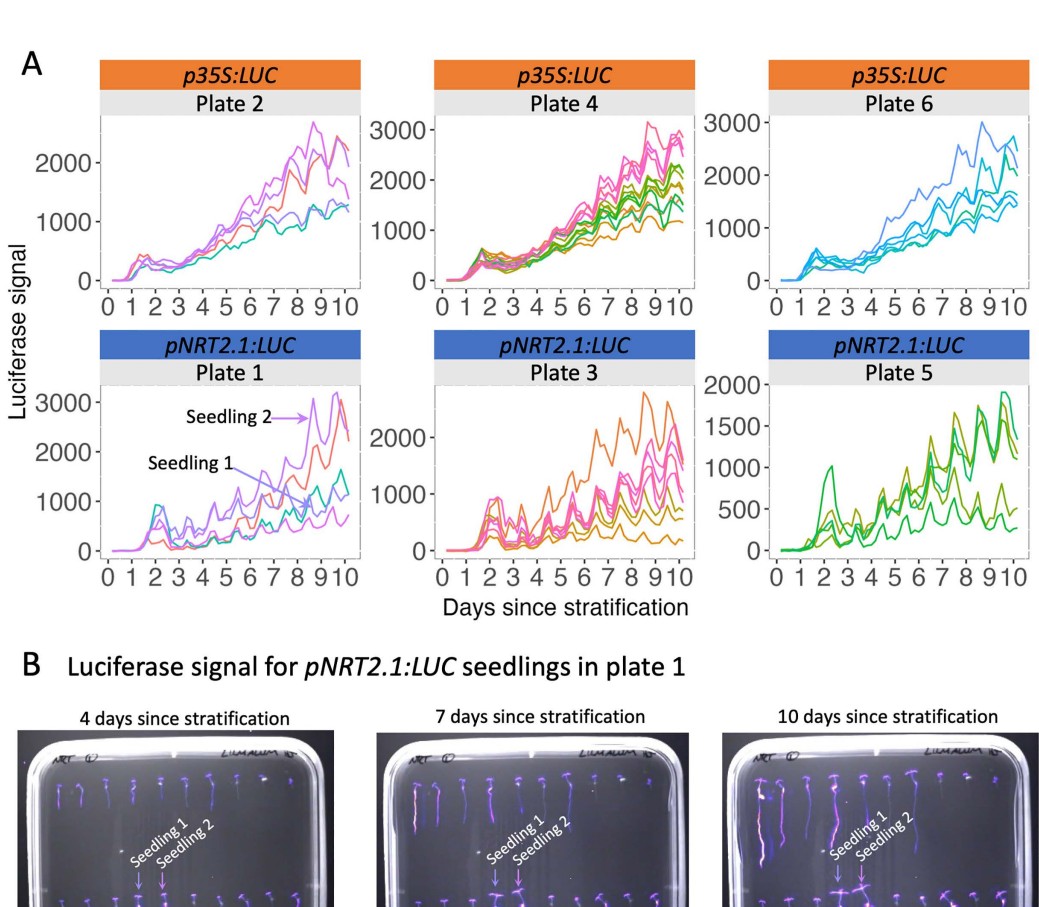

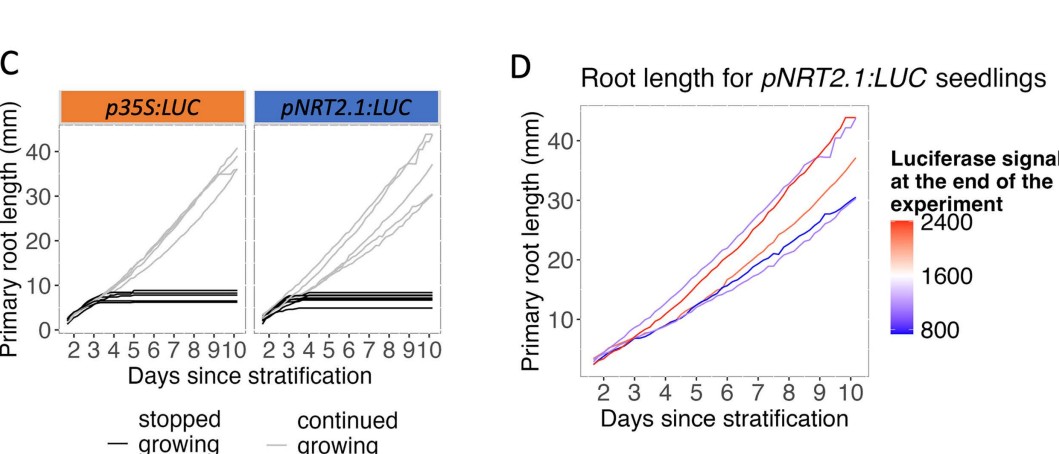

**Fig 3. Differences in *NRT2.1* expression between seedlings are established in young seedlings. (A)** *p35S:LUC* (top) and *pNRT2.1:luc* (bottom) signals over time for 3 plates of each genotype. Five to twelve seedlings were measured in each plate. Only seedlings that had lateral roots at the end of the experiment were measured. Each line represents the signal in a given seedling, measured every 4 hours for 10 days after stratification. **(B)** Images of signal for *pNRT2.1:LUC* seedlings in plate 1 at four, seven and ten days after stratification, overlaid with the bright field images (in grey scale). The colour scale of the luminescence signal is the same in all images and shown on the right. A seedling with low signal (seedling 1) and another one with

a high signal (seedling 2) at the end of the experiment are indicated. **(C)** Primary root length during the time course, measured every 4 hours from one day to ten days after stratification. Seedlings with roots that stop growing and fail to do the heterotroph-to-autotroph transition are shown in dark grey while seedlings with roots that continue to grow throughout the entire experiment are shown in light grey. **(D)** Primary root length during the time course, measured every 4 hours from one day to ten days after stratification for the plants in plate 1 that completed the heterotroph-to-autotroph transition. Lines are coloured based on the normalised luciferase signal for *pNRT2.1:LUC* measured for each seedling at the end of the experiment.

transition [33]. For the other plants, the primary root length continued to grow throughout the experiment, independently of low or high expression of *NRT2.1* (Fig 3B and 3D). This shows that differences in *NRT2.1* expression between seedlings appear shortly after the heterotrophic-to-autotrophic transition.

### *NRT2.1*-relative expression level in plants is maintained for days but is not predictive of *NRT2.1* expression in the next generation

Time-course experiments lasting up to 10 days indicated that relative *NRT2.1* expression in seedlings is maintained despite a nocturnal repression of this gene. To further this study over longer time periods, we then analysed the luciferase signal at 10, 14, 17 and 21 days *in vitro* for *pNRT2.1:LUC* in seedlings grown on low-nitrate conditions. In general, seedlings kept their relative *NRT2.1* expression level throughout the experiment (Fig 4A), leading to a significant positive correlation between the signal of seedlings on the first and last day of the experiment (S4A Fig). This demonstrates that relative *NRT2.1* expression levels in seedlings were maintained for at least up to 21 days. To ensure that these expression differences were not caused by the surrounding microenvironment of the seedlings, we transferred 6-day-old *pNRT2.1:LUC* seedlings to a new plate containing the same medium (low nitrate) and measured their luciferase signal immediately before their transfer, and for 4 days after transfer. Despite their transfer, most seedlings kept their relative expression levels during the experiment (Fig 4B), i.e. seedlings with a high expression before transfer maintained a high expression on new plates and vice-versa. Only a minority of plants drastically changed their expression levels upon transfer, as seen from the positive significant correlation of the luciferase signal in seedlings before and 4 days after transfer (S4B Fig). This demonstrates that differences in *NRT2.1* expression between seedlings were not caused by the microenvironment, but possibly by a state of the seedling that is maintained over several days.

Since relative *NRT2.1* expression levels in seedlings are stable over several days, we next examined whether the relative expression level of a plant could predetermine the expression levels of its descendants. To test this hypothesis, *pNRT2.1:LUC* seedlings with high, medium or low *NRT2.1* expression (Figs 4C, S4C) were transferred to soil for seed production. We then analysed the luciferase signal in their descendants and found similar *NRT2.1* expression levels in all populations, irrespective of the expression levels in their parents (Fig 4C). An independent experiment based on different parents led to an identical outcome (S4C Fig). This result indicates that the relative *NRT2.1* expression level of seedlings is not predictive of the expression level of that gene in the next generation. Furthermore, it suggests that the regulation of *NRT2.1* could reset during the later stages of plant growth (while growing on soil or during seed production). Nevertheless, differences in expression between seedlings are re-established at each generation.

### *NRT2.1* expression level in seedlings is not correlated with its known regulators

To better understand the cause of the inter-individual transcriptional variability for *NRT2.1*, we conducted a transcriptomics study using RNA-seq on 24 individual seedlings grown on a low-nitrate concentration (1 mM $KNO_3$) media. Comparing the transcriptome of each seedling can allow us to measure the level of inter-individual transcriptional variability ($CV^2$) for all genes [9]. By focussing on the genes involved in nitrate acquisition (transport and assimilation; S1 Table), the highly variable status of *NRT2.1* was first confirmed (Fig 5A). Among the genes known to be involved in the regulation of NRT2.1, one protein regulator (PP2C, a type 2C protein phosphatase) and several negative transcriptional regulators (*BT1*, *BT2*,

**A** Analysis of *NRT2.1:LUC* signal for older plants

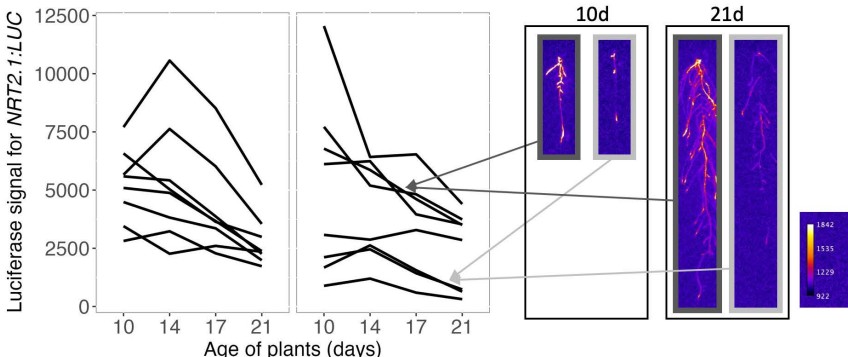

**B** Analysis of *NRT2.1:luc* signal in response to a transfer in a new plate

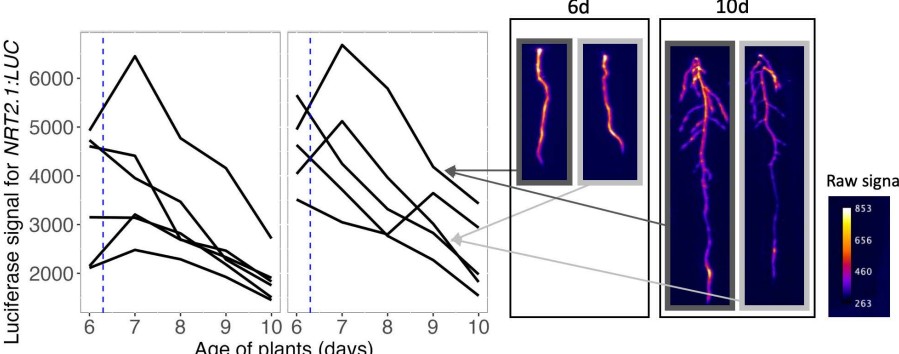

**C** Analysis of *NRT2.1:LUC* signal in parents and offsprings

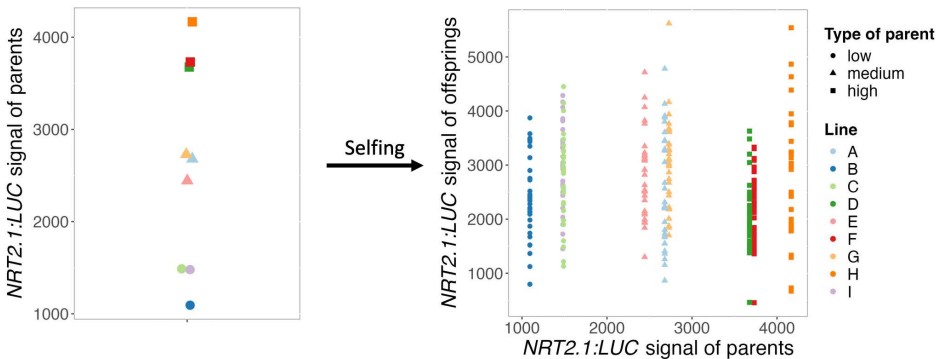

**Fig 4. *NRT2.1* relative expression levels are stable over time but do not influence the signal in the next generation. (A)** *pNRT2.1:LUC* signal over 11 days on older plants for two representative plates. Each line corresponds to values for a given seedling. Corresponding images at 10 and 21 days for seedlings with a low (light grey arrows) or a high (dark grey arrows) *pNRT2.1:LUC* signal are shown on the right. **(B)** *pNRT2.1:LUC* signal over 5 days, after transfer to a new plate, for two plates. Each line corresponds to a given seedling. The vertical blue dashed line indicates the transfer of seedlings to a new plate. Images at 6 days (before transfer) and 10 days for seedlings with a low (light grey highlight) or a high (dark grey highlight) *pNRT2.1:LUC* signal are shown on the right. **(C)** Relation of the *NRT2.1* signal between parents and offsprings. Left: *pNRT2.1:LUC* signal for 9 seedlings. The individuals selected to analyse the signal of their offsprings are shown with different shapes depending on their category: low expression as circles, medium expression as triangles, and high expression as squares. Right: distribution of the *pNRT2.1:LUC* signal for populations derived from self-pollination of the parents selected. Each point corresponds to an individual and the different colours to a population of descendants, with the shape depending on the category of the parent: low expression as circles, medium expression as triangles, and high expression as squares.

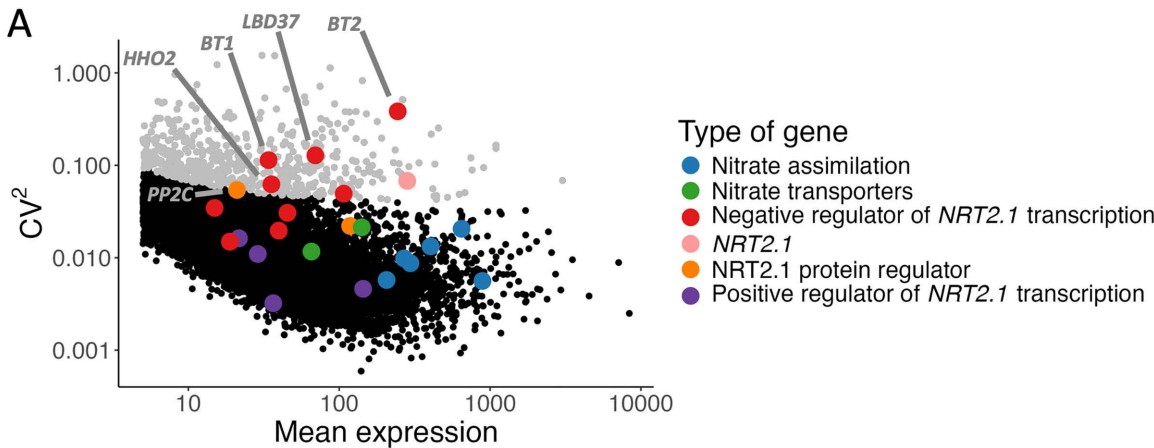

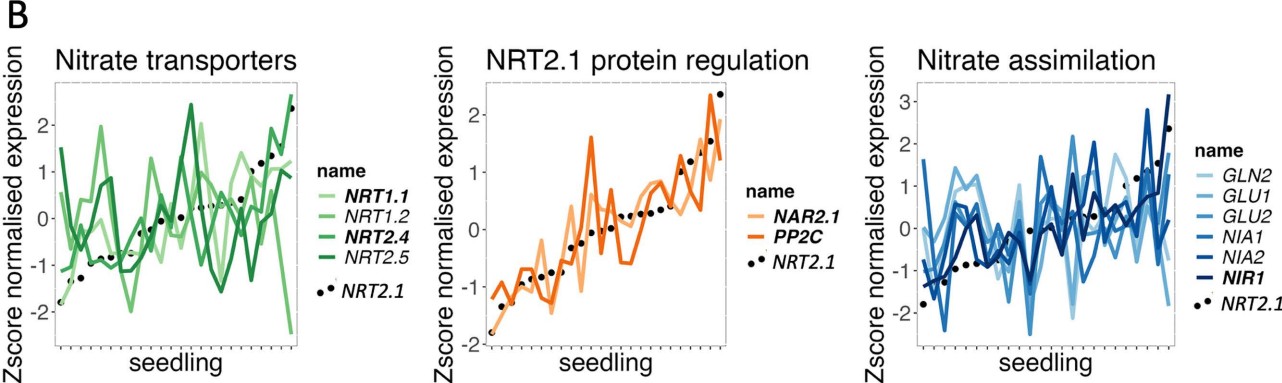

**Fig 5. *NRT2.1* inter-individual transcriptional variability is not associated with its transcriptional regulators. (A)** Representation of the entire transcriptome of the $CV^2$ (variance/mean$^2$) against the mean expression, measured on 24 seedlings. Each point represents one gene, and the points in grey indicate statistically highly variable genes. The genes involved in different aspects of nitrate acquisition are shown in different colours. **(B)** Expression level in the different seedlings of *NRT2.1* (black dots) and nitrate transporters (shades of green, left), genes involved in the post-translational regulation of *NRT2.1* (shades of orange, middle) or nitrate assimilation (shades of blue, right). Genes with a statistically significant correlation are in bold.

*HHO3*, *HHO2* and *LBD37*) possessed particularly high $CV^2$ values (Fig 5A, S1 Table) [34–37]. In contrast, none of the positive transcriptional regulators of *NRT2.1* or genes involved in nitrate assimilation were highly variable (Fig 5A, S1 Table). This result suggests that *NRT2.1* variability could be due to the high inter-individual variability of some of the genes involved in its regulation.

To test this, we compared the expression levels of *NRT2.1* and its regulators on a plant to plant basis to determine if they were correlated (Figs 5B and S5A, S2 Table). In the different seedlings, two genes involved in the regulation of NRT2.1 protein activity, *NAR2.1* and *PP2C* (a type 2C protein phosphatase) [34,38,39], were expressed in a very similar way as *NRT2.1* (p-value <0.05 and $R^2 > 0.7$, Spearman correlation test). Concerning the other gene categories (nitrate transporters, nitrate assimilation, positive or negative transcriptional regulators of *NRT2.1*), we did not observe such a positive correlation in expression between *NRT2.1* and the other genes of each category. However, a subset of genes from some of these categories had an expression very similar to *NRT2.1*, including *NRT2.4* and *NRT1.1* (which encode nitrate transporters) as well as the nitrite reductase-encoding gene *NIR1* (p-value <0.05 and $R^2 > 0.7$, Spearman correlation test). In particular, we did not find a strong correlation between *NRT2.1* and the genes

involved in its transcriptional regulation (S5A Fig, S2 Table). Another transcription factor known to regulate *NRT2.1*, NLP7, was not analysed in this study. Because this transcription factor is not regulated transcriptionally but at the post-translational level [40], we decided to compare the expression in seedlings between *NRT2.1* and the targets of NLP7 and did not find any correlation (S5B Fig). These results suggest that the inter-individual transcriptional variability in *NRT2.1* might not be caused by changes in expression between seedlings for the genes already known to regulate its expression.

**New factors involved in regulating the *NRT2.1* expression level in seedlings**

To find other pathways that might be at the origin of *NRT2.1* inter-individual transcriptional variability, we performed a hierarchical clustering of all genes that vary sufficiently in expression between seedlings (corrected $CV^2 > 1$, measured using the expression in all seedlings). We identified 6 clusters of genes distinguishing specific expression profile trends in the seedlings (Fig 6A, S3 Table). *NRT2.1* falls within cluster 1 (391 genes), and the comparison of the average expression of genes in this cluster with *NRT2.1* demonstrated they closely followed similar trends (Fig 6B). Reciprocally, most genes from this cluster were positively correlated with *NRT2.1* expression (Fig 6C). On the other hand, many of the genes in cluster 3 and cluster 6 were negatively correlated with *NRT2.1*, while most genes in the other clusters were not correlated with *NRT2.1* (S6A, S6B Fig). We then performed a gene ontology (GO) enrichment for the genes in each cluster (S7 Fig). No GO enrichment was found in clusters 3 and 6, whereas we mainly observed in cluster 1 an enrichment for terms related to light and photosynthesis, cell wall and cell component biogenesis and organisation (Fig 6D). Due to the intricate coordination of nitrate and carbon metabolism in plants [41] and to the influence of very high or low levels of nitrogen in the environment on many cell wall-related gene expression [42,43], we wanted to explore the influences of the carbon/nitrogen (C/N) balance and cell wall regulations on *NRT2.1* expression in our data. For this, we analysed the correlation between *NRT2.1* expression in single seedlings and all genes known to be involved in six different energy-associated pathways as a proxy for carbon level regulation [44] as well as previously identified cell wall-related genes [45]. Many of these genes were not in the hierarchical clustering analysis, as their corrected $CV^2$ was lower than 1. Genes involved in photosynthesis represented a high percentage (48%) of genes whose expression was statistically correlated with *NRT2.1* expression in single seedlings (Fig 6E). This correlation was mostly positive rather than negative. This percentage was higher than for genes involved in nitrate acquisition (32%), among which *NAR2.1* and *PP2C* (two regulators of NRT2.1 transporter activity) showed the strongest correlation. No transcriptional regulators of *NRT2.1* were correlated with *NRT2.1* expression in single seedlings. A small percentage of genes involved in ATP biosynthesis, the TCA cycle, and the Asp family pathway correlated with *NRT2.1* expression (Fig 6E). We found the cell wall-related genes had the highest number of genes correlated with *NRT2.1*, both positively and negatively, which can be explained by the fact that this category contained at least twenty times more genes than any other category studied (Fig 6E). In the future, creating sub-categories of cell wall-related genes based on differences in their role might help to understand better which cell-wall related functions are correlated with *NRT2.1*.

These results are difficult to interpret, as they could either mean that: (1) expression of these genes influences *NRT2.1* expression levels in the seedlings; (2) it is actually *NRT2.1* expression that influences them; or (3) this is a correlation with no causality. We thus performed an analysis using the DIANE R package to infer regulatory networks centred around transcription factors to identify the correlation underlying a causality [46]. Using only genes that vary sufficiently in expression between seedlings (corrected $CV^2 > 1$), we inferred a network containing connections between transcription factors and their potential target genes (Fig 7A). The only transcription factor connected to *NRT2.1* in this network was *UNE10* (AT4G00050), also known as *PIF8*. This means *PIF8* is the only transcription factor with a statistically significant correlation with *NRT2.1* to be included in the regulatory network. This indicates that *PIF8* might regulate *NRT2.1* expression, so that it would need to be tested experimentally. No study has explored the targets of this transcription factor, and its binding on *NRT2.1* promoter would have to be confirmed.

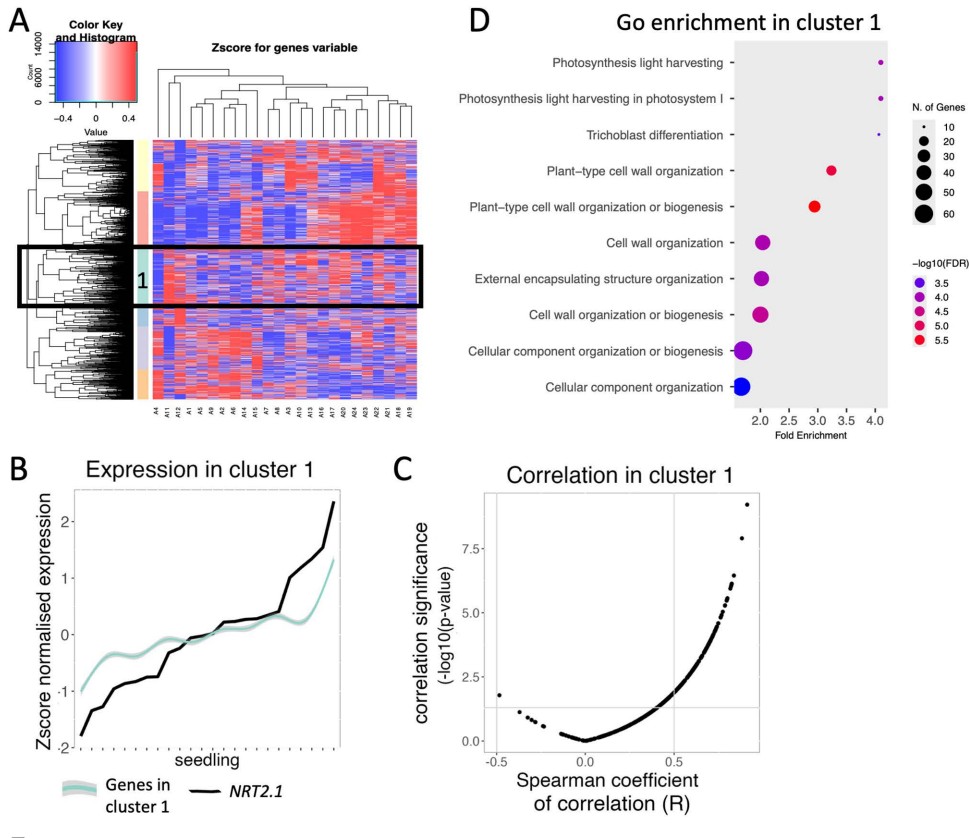

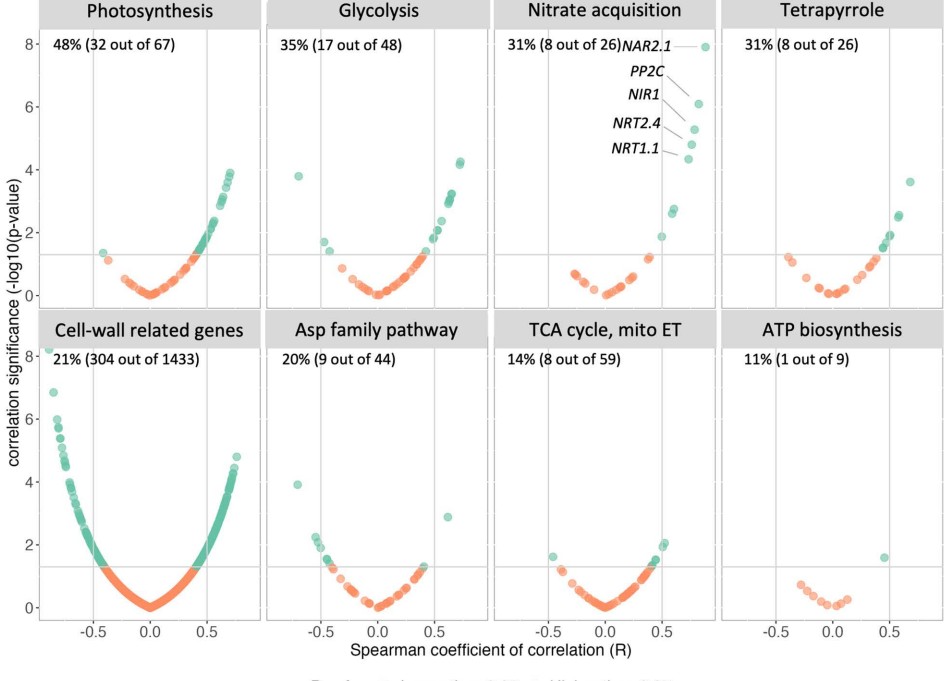

**Fig 6. *NRT2.1* inter-individual transcriptional variability appears to be associated with primary metabolism and cell wall-related genes. (A)** Hierarchical clustering in the different seedlings of genes with a corrected $CV^2 > 1$. The corrected $CV^2$ was measured as described in [9]. The Z-score is

used to correct for expression and variance levels. *NRT2.1* is in cluster 1 (turquoise cluster). **(B)** Expression in the different seedlings of *NRT2.1* (black lines), and of the average and standard deviation of the genes in cluster 1. **(C)** Volcano plots, comparing the correlation significance ($-\log_{10}$(p-value)) with the Spearman coefficient of correlation (R) for the correlation with *NRT2.1* expression in single seedlings for the genes in cluster 1. **(D)** GO enrichment analysis for cluster 1, which contains *NRT2.1*. The top 10 enriched gene ontologies are represented. **(E)** Volcano plots, comparing the correlation significance ($-\log_{10}$(p-value)) with the Spearman coefficient of correlation (R) for the correlation with *NRT2.1* expression in single seedlings, with one plot per category of gene: photosynthesis, glycolysis, nitrate acquisition, tetrapyrrole, cell wall-related genes, Asp pathway, TCA cycle, and ATP biosynthesis. Genes with a p-value < 0.05 are shown in green. The plots for the different categories of genes are ordered depending on the percentage of genes statistically correlated with *NRT2.1* (p-value < 0.05).

### Regulatory network for genes with corrected $CV^2 > 1$

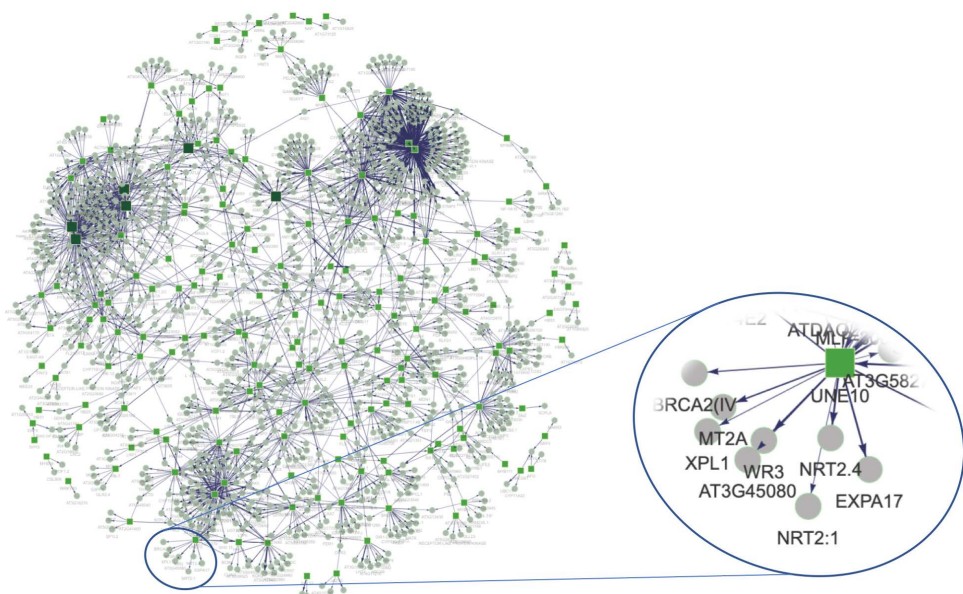

**Fig 7. Gene regulatory network inferred from single seedlings RNA-seq finds new transcription factors potentially regulating *NRT2.1* expression.** Regulatory network inferred using DIANE R package using expression in single seedlings for genes with a corrected $CV^2 > 1$. Edges in the network are represented as arrows going from transcription factors to genes. Transcription factors are indicated as green squares and genes as grey circles. A close-up of the area of the network containing *NRT2.1* is also shown on the right.

## Discussion

We found that *NRT2.1* gene expression is highly variable between genetically identical plants grown in the same environmental condition (Fig 1). The differences in *NRT2.1* promoter activity between plants correlated with the influx of nitrate by a high affinity transporter system, as well as primary root length and lateral root density (Fig 1). We found that *NRT2.1* expression diverges between plants in young seedlings after the heterotrophic-to-autotrophic transition (Fig 3). Furthermore, whereas the relative expression level of *NRT2.1* in seedlings is usually maintained for up to 3 weeks, it is not associated with the *NRT2.1* expression level in the next generation (Fig 4). This led us to wonder what the source of this high inter-individual variability could be. To explore it, we analysed genes that are co-expressed with *NRT2.1* on the genome-wide scale. While known transcriptional regulators of *NRT2.1* were not co-expressed with it, we observed an enrichment of functions related to cell wall-organization and photosynthesis among genes co-expressed with *NRT2.1* (Figs 5 and 6). We also identified *UNE10*, a transcription factor that could possibly regulate *NRT2.1* expression in individual seedlings via the inference of a regulatory network (Fig 7).

Not only did *NRT2.1* exhibit a high inter-individual transcriptional variability, there were also considerable differences between independent experiments regarding the level of *NRT2.1* variability. This could suggest that using 20–30 plants is not enough to robustly measure the level of *NRT2.1* variability. Unfortunately, most studies measure inter-individual variability on a lower number of plants, sometimes with as little as 3–5 plants. Our study also shows the need to measure variability on a high number of independent experiments to ensure its robustness. Sadly, this was also rarely done in previously published studies of inter-individual variability. Future studies will therefore require a thorough investigation of the reproducibility in measuring variability, and of the minimum number of plants required as well as the number of independent experiments to do so. Still, our data already show that inter-individual variability studies must rely on the parallel analysis of more than 20–30 distinct plants.

When exploring the possible phenotypic impact of inter-individual transcriptional variability, we found that differences in *NRT2.1* promoter activity between seedlings correlated with differences in high-affinity root nitrate transport activity (Fig 1D). This is in agreement with previous studies performed on plants showing a strong reduction in nitrate influx by high-affinity root nitrate transport in the *nrt2.1* mutant [30,31]. While *NRT2.1* expression levels do not systematically correlate with the amount of NRT2.1 protein as it is subjected to post-translational regulation [47], it is known that *NRT2.1* expression levels are correlated with nitrate influx by high-affinity root nitrate mainly mediated by NRT2.1 [22]. We found that the expression of two genes coding for proteins involved in NRT2.1 post-translational modification (PP2C and NAR2.1) is positively correlated with *NRT2.1* expression in single seedlings (Fig 5). PP2C (AT4G32950), a type 2C protein phosphatase, was shown to dephosphorylate NRT2.1 protein and activate it [34]. Moreover, NRT2.1 function also requires its interaction with the small transmembrane protein NAR2.1 (AT5G50200) [38,39]. The fact that the expression of these two regulators is correlated with *NRT2.1* expression suggests that they are in the same transcriptional regulatory pathway, and that the level of activation of this pathway is different from one plant to another. In this context, it is very likely that differences between seedlings in the expression of *NRT2.1* and its two regulators *PP2C* and *NAR2.1* might be the cause of the phenotypic variability in high-affinity root nitrate transport for plants grown. To test this, we would need to co-analyse *NRT2.1* expression and high-affinity root nitrate transport in plants with a lower transcriptional variability for *PP2C* and *NAR2.1*, for example by expressing them under the control of a constitutive promoter.

We also found a strong positive correlation between *NRT2.1* promoter activity and two root morphological phenotypes: primary root length and density in lateral roots (Fig 1). In addition, we showed that the *nrt2.1* knockout (KO) mutant has a shorter primary root and a lower lateral root density than the WT when analysed in low-nitrate conditions (1 mM $KNO_3$) (Fig 2). These results are coherent with previously reported observations [31], but differ from others [29]. While the study that agrees with ours was conducted without sucrose, the studies with opposite results used a growth medium containing 1% sucrose. Since it is known that the C/N balance has an important role in regulating plant growth [41], the presence or absence of sucrose in the media in these two other studies might explain the opposite results. Altogether, our results indicate that plant-to-plant variability in primary root length and lateral root density are likely partly caused by the differences in *NRT2.1* expression between plants. In order to test this, we would need to reduce or increase *NRT2.1* inter-individual transcriptional variability without affecting its average expression level, to see if this also affects root growth.

In this study, we identified many cell wall-related genes that are co-expressed with *NRT2.1* in single seedlings. Recently, it was found that cell wall-related genes have high expression variability between sepals in *Arabidopsis thaliana* [45]. It was also shown that both nitrate starvation and high nitrate concentration in the soil affect the expression of many cell wall-related genes [42,43]. Moreover, cell wall relaxation during nitrogen starvation facilitates its assimilation [43]. It is also likely that changes in root architecture that happen in response to nitrogen are at least partly mediated via the cell wall dynamics [42]. However, none of the mutations that affect the cell wall structure or composition had an impact on the expression of genes involved in nitrate acquisition, including *NRT2.1*. We can thus speculate that differences in nitrate uptake between seedlings with a relatively low or high expression level might explain the coordinated changes in cell well-related genes (and not vice versa).

We observed that inter-plant differences in *NRT2.1* expression levels are established in young seedlings after the heterotrophic-to-autotrophic transition (Fig 3). At this stage, plants transition from using reserves accumulated during seed development (triacylglycerols and seed storage proteins in *Arabidopsis thaliana*) to producing their own energy via photosynthesis [48,49]. This period is characterized by massive changes in the regulation of the transcriptome, the epigenome and chromatin condensation [50–52]. For example, Samo and colleagues found that transcriptional repression mediated by Polycomb Repressive Complex 2 (PRC2), via H3K27me3 deposition, facilitates the heterotrophic-to-autotrophic transition and coordinates the metabolic and developmental changes [52]. It is possible that the regulation of genes involved in nitrate acquisition is coordinated with carbon assimilation during this transition. This is supported by previous studies showing that carbon and nitrate metabolism influence each other [53–57]. In agreement with this, our transcriptome analysis shows that genes involved in photosynthesis, nitrate assimilation, and NRT2.1 protein regulation (but not its transcriptional regulation) are strongly co-expressed with *NRT2.1* in seedlings (Figs 5 and 6). Together, this suggests the coordinated regulation of sets of genes, allowing a good C/N balance in plants when photosynthetic activity is established. It also seems that once the differences in expression of *NRT2.1* and genes involved in nitrate and carbon metabolism are established among individual seedlings within a population, their respective differences in expression levels are maintained over time.

None of the known transcriptional regulators of *NRT2.1* appeared in our study [35–37]. As *NRT2.1* is co-expressed with *NAR2.1* and *PP2C*, two genes involved in the regulation of NRT2.1 protein activity, we could suspect these three genes are part of the same regulatory unit, suggesting that some transcription factors that regulate *NRT2.1* expression are still not known. This could be explained by the fact that all transcriptional regulators of *NRT2.1* were identified in studies using strong changes in environmental conditions such as nitrate starvation or the primary nitrate response [19]. Since all plants face the same (or very similar) environment in our analysis, it might allow us to identify transcription factors involved in a finer regulation of *NRT2.1* that would have been masked in previous studies. In this respect, we identified UNE10, also known as PIF8, as an interesting candidate for further study. While no PIFs have been shown to regulate *NRT2.1*, PIF4 represses the expression of *NIA2*, a key gene in nitrate assimilation [58]. Moreover, PIF8 is one of the least studied PIFs, with no available ChIP-seq or DAP-seq data nor any transcriptome of the *pif8* mutant in *Arabidopsis thaliana*. It will thus be interesting to check whether PIF8 binds to the *NRT2.1* promoter, and to quantify *NRT2.1* expression and inter-individual variability in *pif8* mutants. Our results could also suggest that *NRT2.1* high inter-individual transcriptional variability is influenced by other factors such as the TATA box or chromatin modifications, which were shown to regulate variability in expression in other organisms [13].

To determine whether *NRT2.1* is a peculiar case or if other significant differences in gene expression between plants occur shortly after the heterotrophic-to-autotrophic transition, we will need to study the dynamics of expression at the single-plant resolution in young seedlings for other genes, especially those involved in very different functions such as response to stress or development. Auto-bioluminescent reporters with multiple colours have been developed recently in plants [59,60], which might help follow the promoter activity of multiple genes in the same plant. To understand how differences in expression are established between young seedlings, we will also need to explore in more detail the concomitant regulation of transcription and chromatin modifications, since this link has only been studied using pools of seedlings [50–52]. While pooled data improved our knowledge of the transcriptomic and epigenomic dynamics that occur during the heterotrophic-to-autotrophic transition, they might mask plant-to-plant differences, as well as early or subtle regulations. Analysing the transcriptomes and epigenomes of single seedlings before and after the heterotrophic-to-autotrophic transition will allow us to determine if there is any coordinated regulation of genes sets, and the possible role of chromatin in establishing inter-individual transcriptional variability.

To conclude, using single seedlings grown in the same environment, we uncovered new findings about the regulation of *NRT2.1* expression that were masked in previous studies based on the study of pools of plants. Our results indicate that this regulation occurs shortly after the heterotrophic-to-autotrophic transition and is maintained afterwards, and that it is

probably associated with the C/N balance. We also identified a new transcription factor of interest that was overlooked in previous studies focussed on the response to nutritional changes in the soil. Our work showcases the power of studying inter-individual variability using single-plant approaches and how it can improve our understanding of the mechanisms controlling a phenotype of interest.

## Materials and methods

### Plant material and growth conditions

All lines used in this study are in the Col-0 background. The previously published *pNRT2.1:LUC* and *p35S:LUC* lines [24,25] contain 1200 bp of *NRT2.1* promoter and the 35S promoter fused with the luciferase gene, respectively. For the *nrt2.1* KO mutant, we used the SALK_035429 T-DNA insertion line.

Plants were grown on Petri dishes after seed sterilisation and stratification, on solid media containing 1 mM $KH_2PO_4$, 1 mM $MgSO_4$, 0.25 mM $K_2SO_4$, 0.25 mM $CaCl_2$, 0.1 mM Na-Fe-EDTA, 2.5 mM MES and microelements (50 μM KCl, 30 μM $H_3BO_3$, 5 μM $MnSO_4$, 1 μM $ZnSO_4$, 1 μM $CuSO_4$ and 0.1 μM $(NH_4)_6 Mo_7O_{24}$) adjusted at pH 5.8, with 0.8% agar. For experiments with plants up to 13 days old, 12 cm x 12 cm square Petri dishes were used. For experiments on older plants, 23 cm x 23 cm square Petri dishes were used. Only the concentration of $KNO_3$ differed depending on the experiment: 1 mM $KNO_3$ (low nitrate) or 10 mM $KNO_3$ (high nitrate). Except when specified, a set of 4 plates per genotype and per condition were prepared, with 30 seeds per plate (in 2 rows). After 6 days of growth in long (16h) days at 22°C and a light intensity of 150 μmol.m$^{-2}$.s$^{-1}$, seedlings that completed the heterotrophic-to-autotrophic transition and continued to grow were transferred to a new plate and grown in the same conditions. All luciferase experiments included luciferin at a final concentration of 0.25 mM in the growth media.

Replicates for all experiments consisted in fully independent experiments performed on different dates with independently grown plants.

### Luciferase imaging

Except when specified and for time-course experiments, luciferase imaging was carried out on 9-day-old seedlings. Most of the snapshot experiments were conducted using a cooled Hamamatsu C4880-30-24W CCD camera with a 2 minutes acquisition time following a 5 minutes dark period in order to remove any auto-luminescence caused by photosynthetic activity. The imaging in S2D Fig was performed with a cooled Andor iXon ultra 897 EMCCD Back-illuminated camera with a 20 seconds acquisition time following a 5 minutes period of dark. Imaging was carried out around 8 hours after dawn, when expression for *NRT2.1* is high and stable, as its expression depends on the presence of light [61].

For time-course luciferase imaging over several days following germination, up to 11 plates were imaged in parallel in a dedicated plant growth chamber with controlled light and dark phases (16h day/ 8h dark period, with additional short dark phases for luminescence acquisition) and automated luminescence imaging (Lumalum) [32]. The temperature inside the imaging growth cabinet was maintained constantly at about 25°C ± 0.5°C to limit the impact of temperature on luciferase enzymatic activity. For *NRT2.1* time-course luminescence detection, the light intensity of the customized Heliospectra LED module was adjusted to about 125 μmol m$^{-2}$ s$^{-1}$, with independent LED intensity levels adjusted to 2% for 420 nm, 20% for 450 nm, 40% for 530 nm, 10% for 620 nm, 7% for 660 nm and 2% for 735 nm. Throughout the time-course, luciferase imaging was performed every four hours on all plates, with a 60 seconds (bin2) acquisition time. Reference bright field images were taken with a 200 msec exposition (bin2).

The raw integrated density (sum of the signal for all pixels) for luciferase was measured along the primary root (not along the lateral roots) of each seedling using Fiji [62] and then corrected for the length of that primary root as well as for the background signal of the same plate.

### Nitrate influx experiments

Nitrate influx experiments were performed on 11-day-old plants as described in [61]. The plants were first washed with 0.1 mM $CaSO_4$ for 1 min and then incubated in a complete nutrient solution (pH 5.8) containing 0.2 mM $^{15}NO_3^-$ (atom% $^{15}N$: 99%) for 5 min, and finally rinsed again for 1 min with 0.1 mM $CaSO_4$. The seedlings were dried at 70°C for 48h and analysed for total $^{15}N$ contents using an ANCA-MS system (Europa Scientific, Crewe, UK).

### RNA extraction and RT-qPCR

RNA was extracted from 11-day-old individual seedlings using the MagMAX-96 Total RNA Isolation Kit (Ambion) following the manufacturer's instructions, with the following exceptions: 60 µL of Lysis/Binding solution were used instead of 100 µL, and the elution was performed with 25 µL of elution buffer instead of 50 µL. One microgram of RNA was used for reverse transcription, using the M-MLV Reverse Transcriptase (Invitrogen) following the manufacturer's instructions. qPCRs were performed in 384-well plates with 10 µL reactions using a LightCycler 480 (Roche). The primers used in this study were: CAACTACAACCTCACGCAGC and ACCCTTCTTATTCCTCCGGC for the lowly variable gene (AT2G28810); AGTCGCTGATGTCTTGGGAA and AATCTTCCACAGTCCAGCCA for the highly variable gene (AT1G08930); AACAAGGGCTAACGTGGATG and CTGCTTCTCCTGCTCATTCG for *NRT2.1* (AT1G08090); and GGCCTTGTATAATCCCTGATGAATAAG and AAAGAGATAACAGGAACGGAAACATAGT for *UBIQUITIN10*, used as a control gene for the normalisation of expression.

### RNA-seq and mapping

RNA quality and integrity were assessed using a Qubit Flex Fluorometer (Invitrogen). The generation of libraries and sequencing using a PE150 Illumina system were performed by Novogene (Cambridge, UK) with approximately 20 million reads (of each end) per sample. The quality of reads was assessed using FastQC (www.bioinformatics.babraham.ac.uk/projects/fastqc/), and low quality reads were removed and trimmed using fastp [63]. Mapping of the reads on the *Arabidopsis thaliana* genome (TAIR10) was performed with STAR [64], and read count was extracted using htseq-count [65].

### Data analysis

All RNA-seq and RTqPCR analyses and luciferase imaging were performed using R [66], with scripts accessible in GitHub (see data availability). Read counts for each gene were normalized by the size of the library (CPM) using scTenifoldNet package [67]. Highly variable genes were detected as previously described, in addition to the corrected $CV^2$ for each gene [9]. Hierarchical clustering and the heatmap of genes with a corrected $CV^2 > 1$ were performed on the Z-score ((CPM of a gene - mean expression of that gene in samples)/standard deviation of the expression for that gene in samples) using the function hclust on 1-Pearson correlation. Gene Ontology enrichment analysis was performed using the shinyGO web application [68].

Gene regulatory network inference was performed using the R package DIANE [46] for genes with a corrected $CV^2 > 1$.

### Supporting information

**S1 Fig.** (A) Boxplot of the ratio of the pNRT2.1:LUC signal at low (1 mM) and high (10 mM) nitrate concentrations. Each point corresponds to an independent assay. The dotted line at 1 indicates an absence of difference in signal between low and high nitrate, which is not observed in any of the assays. (B) Inter-individual transcriptional variability of the pNRT2.1:LUC reporter line measured for seedlings grown on media with low (1 mM) of high (10 mM) nitrate concentration. Each point represents an independent assay with around 25 seedlings on low nitrate and 25 seedlings on high nitrate per assay. Points from the same assay are linked with a grey dotted line.
(TIFF)

**S2 Fig.** (A) Comparison for the pNRT2.1:LUC line of the differences in signal between seedlings in one condition (1.4x to 1.9x in red) and between the mean signal of seedlings grown in media with low (1 mM) or high (10 mM) nitrate concentration (1.7x in blue). Each point corresponds to the signal in a single seedling. The result of a Wilcoxon test comparing pNRT2.1:LUC signal at low (1 mM) or high (10 mM) nitrate concentration is also included. (B) Comparison for plants grown in media with low (1 mM) nitrate concentration of the pNRT2.1:LUC signal in the primary root and the lateral root density (number of lateral roots/primary root length). Each point corresponds to a single seedling and each colour to an independent assay. The result of a Spearman correlation test for each assay is included. (C-D) Comparison for plants grown in media with high (10 mM) nitrate concentration of the pNRT2.1:LUC signal and the nitrate influx of HATS. Each point corresponds to a single seedling and each colour to an independent assay. The Spearman correlation test result for each assay is included. Luciferase imaging was performed with (C) a Hamamatsu C4880-30-24W CCD camera for half of the assays and (D) an Andor iXon ultra 897 EMCCD Back-illuminated camera for the other half. (E-F) Comparison for plants grown in media with high (10 mM) nitrate concentration of the pNRT2.1:LUC signal at the primary root and (E) the primary root length or (F) the lateral root density (number of lateral roots/primary root length). Each point corresponds to a single seedling and each colour to an independent assay. The result of a Spearman correlation test for each assay is included. (G-H) Comparison for plants grown in media with (G) low (1 mM) or (H) high (10 mM) nitrate concentration of the p35S:LUC signal at the primary root and the primary root length. Each point corresponds to a single seedling and each colour to an independent assay. The result of a Spearman correlation test for each assay is included. (TIFF)

**S3 Fig. Independent assay of** Fig 3A**. pNRT2.1:LUC (top) p35S:LUC (bottom) signal over time.** Seven to eight seedlings were measured in each plate. Only seedlings that have lateral roots at the end of the experiment were measured. Each line represents the signal in a given seedling, measured every 4 hours for 9 days after stratification. (TIFF)

**S4 Fig.** (A) Correlation of the pNRT2.1:LUC signal in 21-day-old and 10-day-old seedlings, corresponding to the data shown in Fig 4A. Each point corresponds to a seedling. The result of a Spearman correlation test is included. (B) Correlation of the pNRT2.1:LUC signal in seedlings before and after a transfer to a new plate with the same concentration of nitrate, corresponding to the data shown in Fig 4B. Each point corresponds to a seedling. The result of a Spearman correlation test is included. (C) Relation in pNRT2.1:LUC signal between parents and offsprings. Left: pNRT2.1:LUC signal for 10 seedlings (independent assay of Fig 4C). The individuals selected to analyse the signal of their offsprings are shown in shapes depending on their category: low expression as circles, medium expression as triangles, and high expression as squares. Right: distribution of the pNRT2.1:LUC signal for populations deriving from self-pollination of the parents selected. Each point corresponds to an individual and the different colours to a population of descendants, with the shape depending on the category of the parent: low expression as circles, medium expression as triangles, and high expression as squares. (TIFF)

**S5 Fig.** (A) Expression level in the different seedlings of NRT2.1 (black dots), and of negative regulators of NRT2.1 expression (shades of red, left), or positive regulators of NRT2.1 expression (shades of purple, right). Genes with a statistically significant correlation are in bold. (B) Expression in the different seedlings of NRT2.1 (black points), and of the average and standard deviation for genes targeted by NLP7 transcription factor. (TIFF)

**S6 Fig.** (A) Expression in the different seedlings of NRT2.1 (black lines), and of the average and standard deviation for the genes in each cluster identified in the hierarchical clustering shown in Fig 6A, with one plot per cluster. (B) Volcano plots, comparing the correlation significance (-log10(p-value)) with the Spearman coefficient of correlation (R) for the

correlation with NRT2.1 expression in single seedlings, with one plot per cluster identified in the hierarchical clustering shown in Fig 6A.
(TIFF)

**S7 Fig.** (A-C) GO enrichment analysis using genes in (A) cluster 2, (B) cluster 4 and (C) cluster 5 identified in the hierarchical clustering shown in Fig 6A. No GO terms were found to be enriched in other clusters (except cluster 1 shown in Fig 6B).
(TIFF)

**S1 Table. Detail of genes involved in nitrate acquisition and whether they are highly variable or not.** This table supports Fig 5A.
(XLSX)

**S2 Table. Correlation in single seedlings with *NRT2.1* for genes in six different energy-associated pathways as a proxy for carbon level regulation as well as previously identified cell wall-related genes.** This table supports Fig 6E.
(XLSX)

**S3 Table. List of genes in each cluster from Fig 6A.**
(XLSX)

**S4 Table. List of genes in each module from Fig 7.**
(XLSX)

**S1 Movie. Movie for a plate containing *pNRT2.1:LUC* measured every 4 hours for 10 days after stratification.** This movie supports Fig 3A.
(AVI)

**S2 Movie. Movie for a plate containing *p35S:LUC* measured every 4 hours for 10 days after stratification.** This movie supports Fig 3A.
(AVI)

## Acknowledgments

We acknowledge the Histocytology and Plant Cell Imaging platform (PHIV) and the Isotopes Quantifications platform (AQUI) for technical support with luciferase imaging and nitrate influx experiments, respectively. We also thank Tou Cheu Xiong for providing the *p35S:LUC* line and Laurine Mancardi for technical help with the Lumalum experiments. We gratefully acknowledge Brandon Loveall from IMPROVENCE for English proofreading of the manuscript.

## Author contributions

**Conceptualization:** Sandra Cortijo.

**Data curation:** Sandra Cortijo.

**Funding acquisition:** Sandra Cortijo.

**Investigation:** Charlotte Lecuyer, Alexandre Vettor, Cécile Fizames, Hélène Javot, Mona Mazouzi, Marie-Hélène Montané, Sandra Cortijo.

**Supervision:** Sandra Cortijo.

**Visualization:** Charlotte Lecuyer, Alexandre Vettor, Sandra Cortijo.

**Writing – original draft:** Charlotte Lecuyer, Alexandre Vettor, Hélène Javot, Antoine Martin, Sandra Cortijo.

**Writing – review & editing:** Charlotte Lecuyer, Alexandre Vettor, Cécile Fizames, Hélène Javot, Antoine Martin, Mona Mazouzi, Marie-Hélène Montané, Sandra Cortijo.

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
