## [Decision Letter · Decision Letter 0]

15 Jul 2025

PGENETICS-D-25-00647

Establishment, maintenance and consequences of inter-individual variability for NRT2.1

PLOS Genetics

Dear Dr. Cortijo,

Thank you for submitting your manuscript to PLOS Genetics. After careful consideration, we feel that it has merit but does not meet PLOS Genetics's publication criteria as it currently stands. Therefore, we invite you to submit a revised version of the manuscript that addresses the points raised during the review process.

Please submit your revised manuscript within 60 days Sep 13 2025 11:59PM. If you will need more time than this to complete your revisions, please reply to this message or contact the journal office at plosgenetics@plos.org. Please include the following items when submitting your revised manuscript:

We look forward to receiving your revised manuscript.

Kind regards,

Jesse Lasky

Academic Editor

PLOS Genetics

Angela Hancock

Section Editor

PLOS Genetics

Aimée Dudley

Editor-in-Chief

PLOS Genetics

Anne Goriely

Editor-in-Chief

PLOS Genetics

**Additional Editor Comments:**

All three reviews are detailed and give constructive critiques. I will look for a revised manuscript to deal with these critiques, including those about tempering and softening some of the authors' conclusions, and removing statements that are not supported by the data. Furthermore, I will expect the reviewers to be able to access the data.

**Journal Requirements:**

At this stage, the following Authors/Authors require contributions: Charlotte Lecuyer, Alexandre Vettor, Cécile Fizames, Hélène Javot, Antoine Martin, Mona Mazouzi, and Sandra Cortijo. Please ensure that the full contributions of each author are acknowledged in the "Add/Edit/Remove Authors" section of our submission form.

The list of CRediT author contributions may be found here: https://journals.plos.org/plosgenetics/s/authorship#loc-author-contributions

https://journals.plos.org/plosgenetics/s/submission-guidelines#loc-parts-of-a-submission

4) We do not publish any copyright or trademark symbols that usually accompany proprietary names, eg ©,  ®, or TM  (e.g. next to drug or reagent names). Therefore please remove all instances of trademark/copyright symbols throughout the text, including:

- TM on pages: 20, and 21.

5) Please upload all main figures as separate Figure files in .tif or .eps format. For more information about how to convert and format your figure files please see our guidelines: 

6) We have noticed that you have uploaded Supporting Information files, but you have not included a list of legends. Please add a full list of legends for your Supporting Information files after the references list.

7) In the online submission form, you indicated that your data will be submitted to a repository upon acceptance. We strongly recommend all authors deposit their data before acceptance, as the process can be lengthy and hold up publication timelines. Please note that, though access restrictions are acceptable now, your entire minimal dataset will need to be made freely accessible if your manuscript is accepted for publication. This policy applies to all data except where public deposition would breach compliance with the protocol approved by your research ethics board. If you are unable to adhere to our open data policy, please kindly revise your statement to explain your reasoning and we will seek the editor's input on an exemption.

8) Please amend your detailed Financial Disclosure statement. This is published with the article. It must therefore be completed in full sentences and contain the exact wording you wish to be published.

9)  Please ensure that the funders and grant numbers match between the Financial Disclosure field and the Funding Information tab in your submission form. Note that the funders must be provided in the same order in both places as well.  

**Reviewers' comments:**

Reviewer's Responses to Questions

**Comments to the Authors:**

Reviewer #1: The manuscript investigates the inter-individual variation in expression of NRT2.1 in Arabidopsis thaliana individuals. LUC reporter activity is used as a proxy for NRT2.1 expression and most experiments are done in seedlings grown on media. NRT2.1 expression variability is investigated under different nitrate concentrations and followed intergenerationally and intragenerationally. NRT2.1 expression is found to be variable between individuals and changes according to environmental conditions. This variability is further connected to three measured phenotypic traits. The claim of NRT2.1 expression variability is admitted to not be novel by the authors but the behaviors under changing conditions are.

If significant changes are made, including the inclusion of detailed study of similar genes, at least at the RNA-seq analysis level, this paper could be of broader interest to the field. My major reservations and comments are outlined below.

Line-by-line comments:

24 - "not transmitted to next generation" - please see my detailed comments below, this is not what was tested in the experimental setup.

36 - "in this state" - Do not understand what state you are referring to

32-38 - The significance statement should be more specific; it could mention crops (like you do later) or at least the unanswered questions in the field you are trying to address

42 - "Phenotypes and gene expression" - Most consider gene expression to be a phenotype

42 - "Interestingly" - Please remove unnecessary linking words

47 - Remove "they"

52 - What consequences? Positive and negative for what? Too vague.

58 - Expound on the mention of crops, it could be mentioned earlier, presumably it is the rationale behind your study

60 - Does selection happen at the level of the individual? Reference please. There is still significant debate in the field, please remove or justify your opinion.

65 - "inter individual" - there is inconsistence in the spelling of inter-individual throughout the paper

83-98 - The justification for the use of NRT2.1 is a bit longwinded and does not explain why you did not select several genes in the same pathway or other highly variable genes.

101 - "as long as the gene is expressed" - nonsensical, rephrase or omit

113 - "of" instead of "for"

115 - Please justify the use of media for most experiments (outside of nitrate concentration experiments), this seems like a major limitation. You did not compare the expression on soil and media despite transferring plants between the two in the "transgenerational" experiment.

117 - Please state the approximate number of seedlings per plate

140 - Figure S1A is your justification for the methodology used throughout the paper. It should be in the main text. Your 2 replicates have very different Rs - please address. I think at least one more rep is needed, and you need to address the variation in correlation per rep.

162 - I am not convinced by the use of 0.5mM here when 1mM is used everywhere else. Is the "enhanced transcriptional response" not desired otherwise? Is there are reference for this?

168 - Please elaborate on this extrapolation. Is an uncoupling of the 2 the only possible explanation?

186-188 - "Loss of expression" - Are you talking about a complete loss of expression? If so, the loss of variability is agiven. If you are talking about very low expression levels, please clarify and propose hypotheses explaining the loss of variability at very low levels of expression.

194 - Fig. 2A - I am assuming there is no significant difference between the treatments. Please make this clear in the figure, the legend and in-text. It is an important caveat.

232-234 - What does this imply?

240 - Fig. 3 - The decision to group the data by replicates does not reveal the whole picture. What is the variance when you combine all 3 replicates? What is the variance when you compare the combined replicates between treatments? Is the difference in variance significant?

288-300 - You attempt to investigate paternal effects, not transgenerational maintenance of expression levels. I have several issues with these experiments and would need to see it repeated or removed from the manuscript. The transfer to soil is a major conflating variable and you have no control for this and no separate experiment where you describe the assumed correlation between expression on media and soil. What if the act of transfer removes the variance even before selfing? Is there a reason you have separated the 2 experiments in Fig. 4D and S4F? Can you instead plot a scatterplot of all measurements as you have before, when you expected a correlation? You also don’t discuss this potential "memory" or lack thereof in the discussion.

349-391 - You conducted whole genome RNA-seq and did extremely limited analysis on it. There are many more questions related to variability you could have addressed with this data. Please conduct a more detailed analysis.

402-403 - That is not what your experiment says.

416-418 - This is a major limitation and should be addressed earlier.

448-449 - Are you suggesting NRT2.1 expression variability is the only cause behind root length and density variability?

Other general comments:

• To increase clarity and comprehension, I strongly suggest English language editing services. There are a lot of verbose sections with unnecessary linking words and some awkward grammar.

• I think this single gene study could be very successfully expanded by including a few more genes (even without adding more phenotypic impact experiments). It would greatly expand the scope of the conclusions that can be drawn.

• There is no mention of circadian fluctuation in NRT2.1 expression, which is a major argument in the cited Cortijo et al. (2019). Was the imagining time controlled? Were steps taken to randomize the order of imaging of reps?

• The Discussion is very verbose. The paragraphs are too long and poorly organized. Please also discuss the potential epigenetic components contributing to expression variability in more details. A few sentences on the implications of the results would be great.

Reviewer #2: The paper focused on one specific gene’s inter-individual expression variability in the model plant *Arabidopsis thaliana* called NRT2.1. The central question of the paper is: when does inter-individual variability established and what are the consequences of this individuality? It’s quite interesting to see that the individuality establishes at a young stage and maintain over days but not generations. However, the present form needs to be better organized and streamlined to increase the readability. In addition, some major results do not appear to have strong supports.

Major comments:

The authors should leverage existing single-cell data to clarify the nature of NRT2.1 expression variability. Specifically, they could distinguish between two possibilities:

1. High cell-to-cell variability within each individual.

2. Relatively uniform expression among cells within an individual, but high variability between individuals.

If this is not feasible, could you discuss why? The relationship between cell-to-cell variability and inter-individual variability is a fun aspect to pursue.

It appears that the gene with higher variability also exhibit high variation of (unstable)CV between experiment, making it difficult to get conclusion with CV. What’s the potential cause of this? Does this suggest the sample size 24 seedlings is insufficient to capture CV. Similarly for Figure S1C, Within each environment, is the variation of CV among 23 experiments high? Any hypothesis of why and potential implications for the entire manuscript? Thus, I question whether the results from Fig1D and 1F are robust - it seems apparent from 1A that after the decrease of CV upon starvation, the CV is still within the range under normal conditions. I suggest that the authors enhance the robustness of the results and move this section to Supplemental materials. or alternatively, remove this section if the results do not have strong support.

It seems the experiments of figure 1 and 4 are similar. Why don’t you use the data from figure 1 (transfer media) to answer the question regarding the effect of environmental fluctuation in relative expression level? I suspect that the samplings of temporal intervals maybe different? Thus, it would be better to have a clear cartoon diagram for figure 1 and 4 to show the design of experiment and clarify on the number/label of plate and number of seedlings/individuals. I’m not sure I understand why Fig4C represents 19 seedlings?

Minor comments:

For Fig 1A, what’s the specific gene for HVG and LVG?

Line 191 The section titles is a little bit vague in a sense it could also just imply a genotype with higher(lower) expression variability exhibit higher(lower) phenotypic variability without correlated expression and phenotypes. Perhaps replace/remove ‘variability’

Line 288 relative NRT2.1 expression level?

Line 467 - 472: the statement read as if they are contradicted. Perhaps increase → decrease?

FigS1C. What is a replicate? a relevant question is what do you mean by ‘combining 23 experiments’ in Line 154. What are differences between experiments? Could authors clarify a bit in the results section?

Fig 4D caption. the parents selected in (B) → selected in (C)?

Line 321 NRT21 → NRT2.1

Line 360  **R² > 0.7 ?**

Reviewer #3: Lecuyer, Vettor et al. is an interesting study in which the authors have gone in depth to analyze inter-individual variability in the expression of NRT2.1, a nitrate transporter gene, in Arabidopsis seedlings. The analysis is quite exhaustive following a sequential series of experiments which reveal unexpected and interesting findings about the nature of inter-individual variability, such as the establishment of variability in early seedlings but not transmitted to the next generation. Experiments of phenotypic consequences mediated by NRT2.1 provide a great complement to the initial findings. The authors analyse correlation between NRT2.1 expression levels and genes involved in nitrate metabolism and then expand the correlation analysis to genome-wide data showing an enrichment of correlated genes involved in photosynthesis.

I find the study attractive and appealing, and I find it will be of interest to different sectors of the community. The experimental findings are robust and I find the video provided a great example of how we need to adopt new forms of sharing results. The network findings are somewhat limited and I invite the authors to go a bit further in that direction to strengthen the message of their manuscript. I also point out some major and minor issues that I consider should be addressed.

Major comments:

I find that the last sentence of the abstract does not reflect the findings of the manuscript accurately. Please consider rephrasing it.

Section: New factors involved in regulating NRT2.1 expression level in seedlings

I am not convinced with the conclusions of the last results section and I believe the data can be exploited a bit more. In particular, I am not convinced that taking the percentage of genes from each pathway with a correlation with NRT2.1 expression with a p-value below 0.05 is an appropriate measure. Arguably, there seem to be many points close to the 0.05 threshold for photosynthesis related genes. The authors could consider using a volcano plot-like figure (correlation value vs. p-value) and take the percentages from there. That said, the size and diversity of the pathways might also influence this result so the percentage might not be accurate even in this case. Additionally, even if the percentage of genes involved in nitrogen nutrition is smaller, their correlation values seem to be higher.

Are the genes with high correlation found in cluster 1? If they are not in cluster 1, it would imply that the results from the clustering (6A) are not revealing genes that are correlated, which would be a problem. It could be that cluster 1 is grouping highly variable genes regardless of their correlation, for example, but this does not seem to be the case according to Figure S5C, so I feel I am missing something here. If they are in cluster 1, it would imply that the results from GO enrichment (6B) are not that relevant to reveal underlying regulatory pathways.

Additionally, what is the overlap between the list of nitrogen nutrition genes used in figure 7C and Figure 6? I would invite the authors to explore this a bit further to strengthen the conclusions.

I find that the findings of early establishment of NRT2.1 expression variability, reversal after change in conditions but no trans-generational inheritance is very interesting and are essential results of the paper, in my opinion. I was somewhat expecting a bit more discussion in this regard and would invite the authors to do so. I consider the discussion of the paper would benefit greatly.

I think the paper would benefit from an effort to homogenize the figures. I feel there is an excess of diversity in terms of font sizes, color schemes, header colors, etc. This could be subjective, of course, but my comment is intended to make the figures clearer. Lack of consistency exists even within figures, which is unnecessary and does not help in transmitting the information adequately. As an example to make a point, the differences between axis labels, axis label fonts, and headers between figure 5A and figure 5B are striking, when they are supposed to be the same type of figure and show the same type of data. Differences between Figure 5A and Figure S3A are also unnecessary.

There is a mistake in the naming of supplementary figures (e.g. Figure S6 is in reality Figure S5). You might want to reconsider your main vs supplementary figure choice. For example, consider merging current Figures 2 and 3 and including current Figure S5C in Figure 7.

Data availability: Although the authors might have strong reasons not to release the raw data generated for this study, I do not see any strong reasons not to make their code available to be reviewed. Given the importance of bioinformatic code for some of the results of this manuscript I would strongly encourage the authors to make their data available prior to acceptance or point to strong reasons why not to. That said, if there are no strong reasons to withhold the raw data prior to acceptance, please reconsider doing so.

Minor comments:

What is the rationale behind the choice of AT1G08930 as HVG and AT2G28810 as LVG? Was RT-qPCR performed de novo for these genes for this work or are the data from Cortijo et al. 2019? Please specify. Were these three the only genes analyzed? If this is de novo work, how do CV values compare to those from Cortijo et al. 2019. I ask this because it would provide more strength to the choice of these genes. With the current writing, it seems arbitrary when I imagine it is not.

Although I appreciate the concluding sentence in each section, I feel that “all in all” is not the most appropriate. Given that the findings are sequential, it might be nice indeed to have a conclusion in each section. I would suggest something like “Finding:” or “Conclusion:” prior to each concluding sentence that stands out. In the current form, the fact that there are short one or two sentence paragraphs at the end of each section is a middle ground that I don’t find convincing.

Figure 2: I suggest using the same color scheme for B as in C and D. “Primary”. Clarify units for lateral root length.

Noting that NRT2.1’s ID is AT1G08090 would be nice in the main text. Additionally, the expression pattern of NRT2.1 across organs according to TAIR seems to be specific to seedlings and roots. It would be nice to point that out.

Line 344: Please list the genes analyzed.

Figure 6: A bit of more work on this figure would make it much more appealing. There seems to be a detachment between the text and figure 6A. Please make the figure legend fit the in-text description. Please clarify that CV2 is used in this case instead of CV (also in the figure legend). Figures 6B-D: consider adding correlation values or visualize it in a way that this is possible. Using a color code that merges Figure 6A with figures 6B-D would be helpful. For example, it is not clear to me why there are only 2 orange points in Figure 6A, but 4 genes in Figure 6B.

Figure 7B: I also note some detachment between what I see in Figure 7B and what I read in the text. For example, plant-type cell wall organization is not mentioned when it appears to have the highest -log10(FDR) value. The following work, measuring inter-individual variability in Arabidopsis sepals, might be relevant in this regard: Hartasánchez et al. 2023: https://doi.org/10.24072/pcjournal.327.

Lines 433-458: I find this paragraph hard to follow. It seems to be going back and forth and the main message is lost. Consider rephrasing the whole paragraph.

The manuscript would benefit from a stringent grammatical review including figure legends. I point out some of the issues I found below. I suppose there are more that I did not see.

Line 74: “ones”

Line 85: “First of all,” and provide citation for previous study

Line 107: “photosynthesis,”

Line 115: “grown on media”

Line 117: “the plant’s”

Line 118: “(CV), that is,”

Line 137: “could” instead of “can”

Line 138: “the NRT2.1”

Line 145: “described above”

Line 146: “However,”

Line 153: Not clear what “as already published” means here

Line 154: “seedlings” and “seedlings”

Line 160: “to” instead of “in”

Line 162: “concentration, with”

Line 163: “respectively, for this experiment specifically” and “saw”

Lines 164 and 165: “a medium” or “media” without the “a”

Line 166: specify that this is CV, just for completeness

Lines 217, 353, 360: “Spearman”

Line 269: “had” instead of “have”. In general, please check for consistency of verbal tense across the manuscript (e.g. line 315, 325)

Line 288. “Is” instead of “are”. In general, please check for consistency in singular versus plural. Consider using “expression levels in seedlings” instead of “level” but try to be consistent.

Line 309: check parenthesis

Line 318: check full stop

Line 362: “known that the”

Line 400: “influx of nitrate by high a affinity transporter system as well as primary root length and lateral root density”

Line 434: “primary root length and lateral root density”

Line 437: “GO terms associated with” instead of “GO involved in”

Line 441 and 446: “low nitrate conditions” or “a low nitrate condition

Line 444: “these” instead of “the 2”

Line 477: “did not”

Line 485: provide citation here

Line 535: “Petri”

Line 620: “Author contributions”

**Have all data underlying the figures and results presented in the manuscript been provided?**

Reviewer #1: **No: ** The authors mention in Data availability that the data will be made available on GitHub after the article is accepted. This made it impossible to review the underlying data and scripts.

Reviewer #2: **No: ** They intend to reveal the data once accepted

Reviewer #3: **No: ** Data and code are not available to review unfortunately (they say they will be available upon acceptance).

PLOS authors have the option to publish the peer review history of their article (what does this mean? ). If published, this will include your full peer review and any attached files.

**Do you want your identity to be public for this peer review?** For information about this choice, including consent withdrawal, please see our Privacy Policy .

Reviewer #1: No

Reviewer #2: No

Reviewer #3: **Yes: ** Diego A. Hartasánchez

**Figure resubmission:**
---

## [Decision Letter · Decision Letter 1]

9 Nov 2025

PGENETICS-D-25-00647R1

Establishment and maintenance of NRT2.1 inter-individual variability in plants

PLOS Genetics

Dear Dr. Cortijo,

Thank you for submitting your manuscript to PLOS Genetics. After careful consideration, we feel that it has merit but does not fully meet PLOS Genetics's publication criteria as it currently stands. Therefore, we invite you to submit a revised version of the manuscript that addresses the remaining points raised. The reviewers appreciated the revisions made, and identified a few outstanding issues.

Please submit your revised manuscript within by Dec 09 2025 11:59PM. If you will need more time than this to complete your revisions, please reply to this message or contact the journal office at plosgenetics@plos.org. Please include the following items when submitting your revised manuscript:

We look forward to receiving your revised manuscript.

Kind regards,

Jesse Lasky

Academic Editor

PLOS Genetics

Angela Hancock

Section Editor

PLOS Genetics

Aimée Dudley

Editor-in-Chief

PLOS Genetics

Anne Goriely

Editor-in-Chief

PLOS Genetics

**Journal Requirements:**

1) We have noticed that you have uploaded Supporting Information files, but you have not included a complete list of legends. Please add a full list of legends for your Supporting Information files after the references list.

2) In the online submission form, you indicated that your data will be submitted to a repository upon acceptance. We strongly recommend all authors deposit their data before acceptance, as the process can be lengthy and hold up publication timelines. Please note that, though access restrictions are acceptable now, your entire minimal dataset will need to be made freely accessible if your manuscript is accepted for publication. This policy applies to all data except where public deposition would breach compliance with the protocol approved by your research ethics board. If you are unable to adhere to our open data policy, please kindly revise your statement to explain your reasoning and we will seek the editor's input on an exemption.

**Reviewers' comments:**

Reviewer's Responses to Questions

**Comments to the Authors:**

Reviewer #1: The authors have substantially improved the manuscript and adequately addressed all of my concerns.

Reviewer #2: The revised manuscript by Lecuyer et al. represents a substantial improvement over the initial submission, with significant text revisions and reorganizing figures. The revised version now provides a more cohesive narrative on how NRT2.1 variability arises and maintained within, but not across, generations.

I would add that regarding the discussion around the regulator of NRT2.1, the determining factor of expression variability does not have to come directly from upstream transcriptional factors. For example, the cis-regulatory region (some cis motif, such as TATA box, are inherently sensitive to subtle environmental changes and hence noisy), epigenetic factors etc. Some epigenetic factors can regulate expression level within generations but not across as well.

Reviewer #3: The authors have performed a substantial revision of their manuscript, increasing clarity in the main message, addressing limitations of their research, and providing more detail in areas where it was lacking. They have addressed all of my comments and suggestions as well as those of the other reviewers. The manuscript is also much better in terms or readability and in consistency across figures and text. I particularly appreciate the modifications and connection between current Figures 5 and 6. I just have a comment on Figure 7 and a few additional minor comments regarding small details on the text.

Main comment:

I recognize that the network analysis might have been done following a suggestion on my part and I appreciate it, yet, with the addition of Figure 7 and the network analysis, new questions come to mind. If I understand correctly, the same genes for the hierarchical clustering are used for the DIANE network. I also understand that DIANE focuses on transcription factors and the finding of PIF8 is relevant. However, Figure 7B shows correlation between NRT2.1 and genes in module 6. I understand that these reveal co-expression. So, my question would be, what is the overlap between cluster 1 and module 6? Why are the GO enrichment results so different? It could be that module 6 is smaller than cluster 1 and hence has lower statistical power, but in any case, this discussion is currently missing from the manuscript and seems to be important.

At this moment I am not sure what I would recommend. One possibility could be to reduce the DIANE results to a mention of PIF8, discussing the other modules less and reducing this bivalence between hierarchical clustering clusters and network modules. Another possibility could be to compare networks and modules, or at least give reasons as to why the results differ. Additionally, for a reader not familiar with DIANE, it is not clear what is meant in lines 365-368 with “UNE10 is the only TF connected to NRT2.1”. Please be more specific. In that regard as well, a section on network reconstruction with DIANE is missing from the material and methods section. To be clear, I think this addition is nice but the way it is presented now adds more noise than clarity. I think the manuscript would benefit from either reducing or increasing this particular section, although the PIF8 finding and discussion should definitely be kept.

Other minor comments:

Line 213. Define WT as wild type.

Line 333. Gene expression “levels”

Lines 343 & 556. Possible double space?

Line 367. No study “has” explored

Line 394. And “of” the minimum number of plants. - to facilitate reading this sentence

Line 411. NRT2.1 “expression”

Line 411. I find that “when it comes to the control of their expression” is not needed, although I might be missing something here

Line 422. Define KO or “knockout”

Line 426. “a growth medium” or “growth media”

Line 426. Define C/N

Line 466. Cite literature for known transcriptional regulators again (34-37 ?)

Line 475. “or DAP-seq”

Line 591. Capital letter “Pearson”

Line 589. “Z-score” (or Zscore if you prefer, but be consistent across the text and figure legends - 5B and 6 for example)

Line 590. Please correct the expression by adding parenthesis: ((CPM of a gene - mean expression of that gene in samples)/standard deviation of the expression for that gene in samples)

Line 955. “as squares”

Lines 973 and 979. GO “terms”

Figure 6D. “GO enrichment”

**Have all data underlying the figures and results presented in the manuscript been provided?**

Reviewer #1: **No: **

Reviewer #2: **No: ** The link provided does not work (at data.recherche.gouv.fr)

Reviewer #3: Yes

PLOS authors have the option to publish the peer review history of their article (what does this mean? ). If published, this will include your full peer review and any attached files.

**Do you want your identity to be public for this peer review?** For information about this choice, including consent withdrawal, please see our Privacy Policy .

Reviewer #1: No

Reviewer #2: No

Reviewer #3: **Yes: ** Diego A. Hartasánchez

**Figure resubmission:**
---

## [Editor Report · Decision Letter 2]

8 Dec 2025

Dear Dr Cortijo,

We are pleased to inform you that your manuscript entitled "Establishment and maintenance of NRT2.1 inter-individual variability in plants" has been editorially accepted for publication in PLOS Genetics. Congratulations!

Yours sincerely,

Jesse Lasky

Academic Editor

PLOS Genetics

Angela Hancock

Section Editor

PLOS Genetics

Aimée Dudley

Editor-in-Chief

PLOS Genetics

Anne Goriely

Editor-in-Chief

PLOS Genetics

BlueSky: @plos.bsky.social

Comments from the reviewers (if applicable):

**Data Deposition**

http://datadryad.org/submit?journalID=pgenetics&manu=PGENETICS-D-25-00647R2

**Press Queries**

---

## [Editor Report · Acceptance letter]

PGENETICS-D-25-00647R2

Establishment and maintenance of NRT2.1 inter-individual variability in plants

Dear Dr Cortijo,

We are pleased to inform you that your manuscript entitled " 

Establishment and maintenance of NRT2.1 inter-individual variability in plants " has been formally accepted for publication in PLOS Genetics! Your manuscript is now with our production department and you will be notified of the publication date in due course.

With kind regards,

Zsofia Freund

PLOS Genetics

On behalf of:
